# Regret-Optimal Model-Free Reinforcement Learning for Discounted MDPs with Short Burn-In Time

**Xiang Ji** [*]
Princeton

**Gen Li** [†]
CUHK

## Abstract

A crucial problem in reinforcement learning is learning the optimal policy. We study this in tabular infinite-horizon discounted Markov decision processes under the online setting. The existing algorithms either fail to achieve regret optimality or have to incur a high memory and computational cost. In addition, existing optimal algorithms all require a long burn-in time in order to achieve optimal sample efficiency, i.e., their optimality is not guaranteed unless sample size surpasses a high threshold. We address both open problems by introducing a model-free algorithm that employs variance reduction and a novel technique that switches the execution policy in a slow-yet-adaptive manner. This is the first regret-optimal model-free algorithm in the discounted setting, with the additional benefit of a low burn-in time.

## 1 Introduction

In reinforcement learning (RL), a crucial task is to find the optimal policy that maximizes its expected cumulative reward in any given environment with unknown dynamics. An immense body of literature is dedicated to finding algorithms that solve this task with as few samples as possible, which is the prime goal under this task. Ideally, one hopes to find an algorithm with a theoretical guarantee of optimal sample efficiency. At the same time, this task might be accompanied with additional requirements such as low space complexity and computational cost, as it is common that the state and action spaces exhibit high dimensions in modern applications. The combination of these various goals and requirements presents an important yet challenging problem in algorithm design.

The task of searching for optimal policy has been well-studied by existing work in the generative setting [33; 34; 21; 1]. This fundamental setting allows the freedom of querying samples at any state-action pair. In contrast, it is more realistic but difficult to consider the same task in the online setting, in which samples can only be collected along trajectories generated from executing a policy in the unknown Markov decision process (MDP). Solving this task with optimal sample efficiency requires a careful balance between exploration and exploitation, especially when coupled with other goals such as memory and computational efficiency.

MDPs can be divided into two types: the episodic finite-horizon MDPs and the infinite-horizon MDPs. Although these two types of MDPs can be approached in similar ways under the generative setting, there is a clear dichotomy between them in the online setting. In an episodic MDP, sample trajectories are only defined in fixed-length episodes, so samples are collected in episodes, and a reset to an arbitrary initial state occurs at the end of every online episode. Its transition kernel is usually assumed to be non-stationary over time. In contrast, the transition kernel of an infinite-horizon MDP stays stationary over time, and the online sample collection process amounts to drawing a single infinitely long sample trajectory with no reset. These differences render most optimal algorithms for episodic

---

[*]Department of Electrical and Computer Engineering, Princeton University, Princeton, NJ 08544, USA.
[†]Department of Statistics, The Chinese University of Hong Kong, Hong Kong, China.

37th Conference on Neural Information Processing Systems (NeurIPS 2023).

MDPs suboptimal when applied to infinite-horizon MDPs. Without reset and non-stationarity, the high dependency between consecutive trajectory steps in the infinite-horizon setting presents a new challenge over the episodic setting. In this work, we consider the infinite-horizon discounted MDPs, which is widely used in practice but still has some fundamental questions unanswered in theory.

## 1.1 Sample Efficiency in Infinite-Horizon MDPs

To evaluate the sample efficiency of online RL algorithms, a natural and widely-accepted metric is the *cumulative regret*. It captures the performance difference between the optimal policy and the learned policy of an algorithm over its online interactions with a given MDP. The notion of cumulative regret was first introduced in the bandit literature and later adopted in the RL literature [2; 15]. It is profusely used in the online episodic RL literature. Such works aim to prove regret guarantees for algorithms and provide analyses that characterize such regret guarantees in terms of all problem parameters such as state space, action space and sample size in a non-asymptotic fashion. A cumulative regret guarantee can also suggest the sample complexity needed to reach a certain level of average regret.

In the online infinite-horizon setting, many works study a different metric called the sample complexity of exploration, first introduced in [17]. In essence, given a target accuracy level $\epsilon$, this metric characterizes the total number of $\epsilon$-suboptimal steps committed by an algorithm over an infinitely-long trajectory in the MDP. While this is indicative of the sample efficiency of an algorithm, the focus of this metric is very different from that of cumulative regret, as it only reflects the total number of failures but does not distinguish their sizes. As [24; 12] point out, even an optimal guarantee on the sample complexity of exploration can only be converted to a very suboptimal guarantee on the cumulative regret. To obtain a more quantitative characterization of the total volume of failures in the regime of finite samples, some works have turned to studying cumulative regret guarantees for algorithms.

It was not until recently that some works [24; 51; 12; 18] begin to research into the problem of cumulative regret minimization in infinite-horizon discounted MDPs. Among them, [51] focus on linear MDPs while others study tabular MDPs. In this work, we study the regret minimization problem in the tabular case. Hereafter and throughout, we denote the size of the state space, the size of the action space and the discount factor of the problem MDP with $S$, $A$ and $\gamma$, respectively, and let $T$ denote the sample size.

## 1.2 Model-Based and Model-Free Methods

Since modern RL applications are often large-scale, algorithms with low space complexity and computational complexity are much desired. This renders the distinction between model-based algorithms and model-free algorithms particularly important. The procedure of a model-based method includes a model estimation stage that involves estimating the transition kernel and a subsequent planning stage that searches the optimal policy in the learned model. Thus, $O(S^2A)$ space is required to store the estimated model. This is unfavorable when the state space is large and a memory constraint is present. Additionally, updating the transition kernel estimate brings a large computational burden. In comparison, model-free methods do not learn the entire model and thus can run with $o(S^2A)$ space. Notably, most value-based methods such as Q-learning only require storage of an estimated Q-function, which can take as little as $O(SA)$ memory. In the infinite-horizon discounted setting, although UCBVI-$\gamma$ in [12] can achieve optimal regret, its model-based nature exacts a $O(S^2A)$ memory and computational cost; conversely, the algorithms in [24] and [18] are model-free but have suboptimal regret guarantee.

## 1.3 Burn-in Cost in Regret-Optimal RL

Naturally, one aims to develop algorithms that find the optimal policy with the fewest number of samples. In regards to regret, this motivates numerous works to work towards algorithms with minimax-optimal cumulative regret. However, the job is not done once such an algorithm is found. As can be seen in the episodic RL literature, algorithms that achieve optimal regret as sample size $T$ tends towards infinity can still have different performance in the regime when $T$ is limited. Specifically, for every existing algorithm, there exists a certain sample size threshold such that regret is suboptimal before $T$ exceeds it. Such threshold is commonly referred to as the initial *burn-in time* of the algorithm. Therefore, it is of great interest to find an algorithm with low burn-in time so that it

| Algorithm | Sample complexity of exploration | Cumulative Regret | Range of $T$ with optimal regret | Space complexity |
|---|---|---|---|---|
| Delayed Q-learning [35] | $\frac{SA}{(1-\gamma)^8\epsilon^4}$ | $\frac{S^{\frac15}A^{\frac15}T^{\frac45}}{(1-\gamma)^{\frac95}}$ | never | $SA$ |
| R-Max [7] | $\frac{S^2A}{(1-\gamma)^6\epsilon^3}$ | $\frac{S^{\frac12}A^{\frac14}T^{\frac34}}{(1-\gamma)^{\frac74}}$ | never | $S^2A$ |
| UCB-Q [8] | $\frac{SA}{(1-\gamma)^7\epsilon^2}$ | $\frac{S^{\frac13}A^{\frac13}T^{\frac23}}{(1-\gamma)^{\frac83}}$ | never | $SA$ |
| MoRmax [36] | $\frac{SA}{(1-\gamma)^6\epsilon^2}$ | $\frac{S^{\frac13}A^{\frac13}T^{\frac23}}{(1-\gamma)^{\frac73}}$ | never | $S^2A$ |
| UCRL [20] | $\frac{S^2A}{(1-\gamma)^3\epsilon^2}$ | $\frac{S^{\frac23}A^{\frac13}T^{\frac23}}{(1-\gamma)^{\frac43}}$ | never | $S^2A$ |
| UCB-multistage [49] | $\frac{SA}{(1-\gamma)^{\frac{11}{2}}\epsilon^2}$ | $\frac{S^{\frac13}A^{\frac13}T^{\frac23}}{(1-\gamma)^{\frac{13}{6}}}$ | never | $SA$ |
| UCB-multistage-adv[1] [49] | $\frac{SA}{(1-\gamma)^3\epsilon^2}$ | $\frac{S^{\frac13}A^{\frac13}T^{\frac23}}{(1-\gamma)^{\frac43}}$ | never | $SA$ |
| MAIN[2] [18] | N/A | $\kappa\sqrt{\frac{(S^4+S^2A^2)T}{(1-\gamma)^8}}$ | never | $SA$ |
| Double Q-learning [24] | N/A | $\sqrt{\frac{SAT}{(1-\gamma)^5}}$ | never | $SA$ |
| UCBVI-$\gamma$[3] [12] | N/A | $\sqrt{\frac{SAT}{(1-\gamma)^3}}$ | $\left[\frac{S^3A^2}{(1-\gamma)^4},\infty\right)$ | $S^2A$ |
| Q-SlowSwitch-Adv (**This work**) | N/A | $\sqrt{\frac{SAT}{(1-\gamma)^3}}$ | $\left[\frac{SA}{(1-\gamma)^{13}},\infty\right)$ | $SA$ |
| Lower bound [20]; [12] | $\frac{SA}{(1-\gamma)^3\epsilon^2}$ | $\sqrt{\frac{SAT}{(1-\gamma)^3}}$ | N/A | N/A |

Table 1: A comparison between our results and existing work in the online infinite-horizon discounted setting. Logarithmic factors are omitted for clearer presentation. The second column shows the sample complexity when the target accuracy $\epsilon$ is sufficiently small. The third column shows the regret when sample size $T$ is sufficiently large (beyond the burn-in period). The algorithms in the first seven rows only have sample complexity results in their original works; their regret bounds are derived from their respective sample complexity bounds and presented in this table for completeness. Details about the conversions can be found in [12]. The fourth column lists the sample size range in which regret optimality can be attained, which shows the burn-in time. We would like to point out that the results in [12; 24] are under slightly different regret definitions from the regret definition in [18] and this work. In fact, their regret metric can be more lenient, and our algorithm can also achieve $\widetilde{O}(\sqrt{\frac{SAT}{(1-\gamma)^3}})$ optimal regret under it. This is further discussed in Remark 1 and Appendix B.

can still attain optimal regret in the sample-starved regime. Such effort has been made by [21; 1] in the generative setting and by [23; 26] in the online episodic setting. Yet, this important issue has not been addressed in the infinite-horizon setting, as optimal algorithms all suffer long burn-in times.

Specifically, while UCBVI-$\gamma$ in [12] achieves a state-of-the-art regret guarantee of $\widetilde{O}\left(\sqrt{\frac{SAT}{(1-\gamma)^3}}\right)$, which they prove minimax-optimal, their theory does not guarantee optimality unless the samples size $T$ becomes as large as

$$T \geq \frac{S^3A^2}{(1-\gamma)^4}.$$

---

[1]UCB-multistage-adv achieves optimal sample complexity only in the high-accuracy regime when $\epsilon \leq S^{-2}A^{-2}(1-\gamma)^{14}$. This is similar to a burn-in threshold in that the optimal guarantee cannot be achieved unless in a specific range.

[2]The regret analysis for MAIN assumes an ergodicity parameter $\kappa$.

[3]UCBVI-$\gamma$ achieves optimal regret for $T \geq \frac{S^3A^2}{(1-\gamma)^4}$ only if the MDP satisfies $SA \geq \frac{1}{1-\gamma}$.

This threshold can be prohibitively large when $S$ and $A$ are huge, which is true in most applications. Thus, this makes reducing these factors in the burn-in cost particularly important. For instance, a 5-by-5 tic-tac-toe has a state space of size $3^{25}$. While this is a manageable number in modern machine learning, any higher power of it may cause computational difficulties; in contrast, the horizon of this game is much smaller—no more than 25. Since no lower bound precludes regret optimality for $T \geq \frac{SA}{(1-\gamma)^4}$, one might hope to design an algorithm with smaller $S$ and $A$ factors in the burn-in cost so that it can achieve optimality even in the sample-starved regime.

## 1.4 Summary of Contributions

While it is encouraging to see recent works have shown that in the discounted setting, model-free methods can provide nearly optimal guarantees on sample complexity of exploration and that model-based methods can provide nearly optimal finite-sample regret guarantees, there still lacks a *model-free* approach that can attain *regret optimality*. In the orthogonal direction, there is still a vacancy for algorithms that can attain optimal regret for a broader sample size range, i.e., with fewer samples than $\frac{S^3 A^2}{\text{poly}(1-\gamma)}$.

In fact, we can summarize these two lingering theoretical questions as follows:

*Is there an algorithm that can achieve minimax regret optimality with low space complexity and computational complexity in the infinite-horizon discounted setting, even when sample size is limited?*

We answer this question affirmatively with a new algorithm Q-SlowSwitch-Adv, which uses variance reduction and a novel adaptive switching technique. It is the first model-free algorithm that achieves optimal regret in the infinite-horizon discounted setting. This result can be summarized as follows:

**Theorem** (informal). *For any sample size $T \geq \frac{SA}{\text{poly}(1-\gamma)}$, Q-SlowSwitch-Adv is guaranteed to achieve near-optimal cumulative regret $\widetilde{O}\left(\sqrt{\frac{SAT}{(1-\gamma)^3}}\right)$ with space complexity $O(SA)$ and computational complexity $O(T)$.*

A formal theorem is presented in Section 4; its proof can be found in the full version [14]. We also provide a complete summary of related prior results in Table 1. A discussion about the additional related work is deferred to Appendix A.

## 2 Problem Formulation

Let us specify the problem we aim to study in this section. Throughout this paper, we let $\Delta(\mathcal{X})$ denote the probability simplex over any set $\mathcal{X}$. We also introduce the notation $[m] := \{1, 2, \cdots, m\}$ for a positive integer $m$.

### 2.1 Infinite-Horizon Discounted Markov Decision Process

We consider an infinite-horizon discounted Markov decision process (MDP) represented with $(\mathcal{S}, \mathcal{A}, \gamma, P, r)$. Notably, we consider a tabular one, in which $\mathcal{S} := \{1, 2, \cdots, S\}$ denotes the state space with size $S$ and $\mathcal{A} := \{1, 2, \cdots, A\}$ denotes the action space with size $A$. $P : \mathcal{S} \times \mathcal{A} \to \Delta(\mathcal{S})$ denotes the probability transition kernel in that $P(\cdot|s, a) \in \Delta(\mathcal{S})$ is the transition probability vector from state $s \in \mathcal{S}$ when action $a \in \mathcal{A}$ is taken. $r : \mathcal{S} \times \mathcal{A} \to [0, 1]$ denotes the reward function, which is assumed to be deterministic in this work. Specifically, $r(s, a)$ is the immediate reward for taking action $a \in \mathcal{A}$ at state $s \in \mathcal{S}$. Lastly, $\gamma$ denotes the discount factor for the reward, which makes $\frac{1}{1-\gamma}$ the effective horizon.

A (stationary) policy $\pi : \mathcal{S} \to \Delta(\mathcal{A})$ specifies a rule for action selection in that $\pi(\cdot|s) \in \Delta(\mathcal{A})$ is the action selection probability vector at state $s \in \mathcal{S}$. We overload this notation by letting $\pi(s)$ denote the action policy $\pi$ takes at state $s$. Given a policy $\pi$, the Q-function of $\pi$ is defined as

$$Q^\pi(s, a) := \mathbb{E}\left[\sum_{t=0}^\infty \gamma^t r(s_t, a_t) \mid s_0 = s, a_0 = a\right],$$

in which $s_{t+1} \sim P(\cdot|s_t, a_t)$ for $t \geq 0$ and $a_t \sim \pi(\cdot|s_t)$ for $t \geq 1$. Moreover, the value function of $\pi$ is defined as

$$V^\pi(s) := \mathbb{E}\left[\sum_{t=0}^{\infty} \gamma^t r(s_t, a_t) \,\Big|\, s_0 = s\right],$$

in which $s_{t+1} \sim P(\cdot|s_t, a_t)$ and $a_t \sim \pi(\cdot|s_t)$ for $t \geq 0$. The Q-function and value function satisfy an equation, called the Bellman equation [6]:

$$Q^\pi(s, a) = r(s, a) + \gamma \mathbb{E}_{s' \sim P(\cdot|s,a)}\left[V^\pi(s')\right]. \tag{1}$$

A policy $\pi^\star$ is called an optimal policy if it maximizes the value function for all states simultaneously. The optimal value function and optimal Q-function can be defined as

$$V^\star(s) := \max_\pi V^\pi(s) = V^{\pi^\star}(s)$$

$$Q^\star(s, a) := \max_\pi Q^\pi(s, a) = Q^{\pi^\star}(s, a),$$

which satisfy

$$V^\star(s) = V^{\pi^\star}(s) \quad \text{and} \quad Q^\star(s, a) = Q^{\pi^\star}(s, a)$$

for any optimal policy $\pi^\star$. The optimal policy always exists and satisfies the Bellman optimality equation [30]:

$$Q^{\pi^\star}(s, a) = r(s, a) + \gamma \mathbb{E}_{s' \sim P(\cdot|s,a)}\left[\max_{a' \in \mathcal{A}} Q^{\pi^\star}(s', a')\right]$$

$$= r(s, a) + \gamma \mathbb{E}_{s' \sim P(\cdot|s,a)}\left[V^\star(s')\right]. \tag{2}$$

## 2.2 Online Learning in an Infinite-Horizon MDP

We consider the online (single-trajectory) setting, in which the agent is permitted to execute a total of $T$ steps sequentially in the MDP. More specifically, the agent starts from an arbitrary (and possibly adversarial) initial state $s_1$. At each step $t \in [T]$, the agent at state $s_t$ computes policy $\pi_t$, takes action $a_t$ based on $\pi_t(\cdot|s_t)$, receives reward $r(s_t, a_t)$, and transitions to state $s_{t+1}$ in the following step. At the end of execution, the agent generates a trajectory $(s_1, a_1, r_1, s_2, a_2, r_2, \cdots, s_T, a_T, r_T)$, which amounts to $T$ samples.

## 2.3 Problem: Regret Minimization

As a standard metric to evaluate the performance of the aforementioned agent over a finite number of $T$ steps, the cumulative regret with respect to the sequence of stationary policies $\{\pi_t\}_{t=1}^T$ learned by the algorithm is defined as follows:

$$\text{Regret}(T) := \sum_{t=1}^{T}\left(V^\star(s_t) - V^{\pi_t}(s_t)\right). \tag{3}$$

Verbally, the regret measures the cumulative suboptimality between the optimal policy and the execution policy $\pi_t$ at each step throughout the $T$-step online interaction process. Naturally, one aims to minimize this regret by finding an algorithm whose regret scales optimally in $T$. This would require a strategic balance between exploration and exploitation, which can be difficult when sample size $T$ is small.

***Remark*** 1. In the infinite-horizon setting, many prior works [12; 24] consider slightly different regret definitions with respect to non-stationary policies. Specifically, at each $s_t$ along the trajectory, this different regret metric compares the optimal value function $V^\star(s_t)$ against the expected cumulative reward of running the non-stationary policy $\{\pi_k\}_{k=t}^\infty$ starting from $s_t$. By doing this, it is effectively evaluating the cumulative reward difference between the stationary optimal policy and a non-stationary algorithm. While there exists no formal conversion between the regret defined in this way and the one in (3), it is expected to be smaller and thus more easily controlled than (3), because the execution policy $\pi_t$ improves over time. In addition, we can show our algorithm also achieves the same level of

---

**Algorithm 1:** Q-SlowSwitch-Adv

---

1 **Initialize:** $\forall(s,a), Q^{\text{lazy}}(s,a), Q(s,a), Q^{\text{UCB}}(s,a), Q^{\text{R}}(s,a), Q_M(s,a) \leftarrow \frac{1}{1-\gamma}$;
  $N(s,a) \leftarrow 0; V(s), V^{\text{R}}(s) \leftarrow \frac{1}{1-\gamma}; Q^{\text{LCB}}(s,a), \theta(s,a) \leftarrow 0; \mathcal{D} \leftarrow \text{dict}()$;
  $\mu^{\text{ref}}(s,a), \sigma^{\text{ref}}(s,a), \mu^{\text{adv}}(s,a), \sigma^{\text{adv}}(s,a), B^{\text{R}}(s,a), \delta^{\text{R}}(s,a) \leftarrow 0; u^{\text{switch}} \leftarrow \text{False}$;
  $u^{\text{ref}}(s) \leftarrow \text{True}; H = \lceil \frac{2}{1-\gamma} \rceil; \iota = \log \frac{SAT}{\delta}$.
2 **for** $t = 1, \cdots, T$ **do**
3      Take action $a_t = \pi_t(s_t) = \arg\max_a Q^{\text{lazy}}(s_t, a)$, and draw $s_{t+1} \sim P(\cdot|s_t, a_t)$;
4      $N(s_t, a_t) \leftarrow N(s_t, a_t) + 1; n \leftarrow N(s_t, a_t)$;   # Update the counter
5      $\eta_n \leftarrow \frac{H+1}{H+n}$;   # Update the learning rate
6      $Q^{\text{UCB}}(s_t, a_t) \leftarrow$ **update-q-ucb**();   # Compute the UCB. See Algorithm 2
7      $Q^{\text{LCB}}(s_t, a_t) \leftarrow$ **update-q-lcb**();   # Compute the LCB. See Algorithm 2
8      $Q^{\text{R}}(s_t, a_t) \leftarrow$ **update-q-reference**();   # Compute the reference value. See Algorithm 2
9      $Q(s_t, a_t) \leftarrow \min\{Q^{\text{R}}(s_t, a_t), Q^{\text{UCB}}(s_t, a_t), Q(s_t, a_t)\}$;
10      $V(s_t) \leftarrow \max_a Q(s_t, a)$;
11      $V^{\text{LCB}}(s_t) \leftarrow \max\{\max_a Q^{\text{LCB}}(s_t, a), V^{\text{LCB}}(s_t)\}$;
12      $\theta(s_t, a_t) \leftarrow \theta(s_t, a_t) + Q_M(s_t, a_t) - Q(s_t, a_t)$.   # Track the staleness of current policy
13      **if** $u^{\text{switch}} = \text{True}$ **then**
14          $Q^{\text{lazy}} \leftarrow$ **update-q-lazy**();   # Update execution policy's Q-function (switch policy)
15          $\mathcal{D} \leftarrow \text{dict}()$;   # Reset the buffer
16          $u^{\text{switch}} \leftarrow \text{False}$;
17      $\mathcal{D}[(s_t, a_t)] \leftarrow Q(s_t, a_t)$.   # Add the new transition and $Q$ entry to the buffer
18      """Switch policy when the staleness tracker is large"""
19      **if** $\theta(s_t, a_t) > \frac{1}{1-\gamma}$ **then**
20          $Q_M(s_t, a_t) \leftarrow Q(s_t, a_t)$;   # Save the current $Q$ entry for staleness determination later
21          $u^{\text{switch}} \leftarrow \text{True}$;   # Signal to switch policy at the following step
22          $\theta(s_t, a_t) \leftarrow 0$;   # Reset the staleness tracker
23      **if** $V(s_t) - V^{\text{LCB}}(s_t) > 3$ **then**
24          $V^{\text{R}}(s_t) \leftarrow V(s_t), u^{\text{ref}}(s_t) = \text{True}$;
25      **else if** $u^{\text{ref}}(s_t) = \text{True}$ **then**
26          $V^{\text{R}}(s_t) \leftarrow V(s_t), u^{\text{ref}}(s_t) = \text{False}$.   # Update the reference only on certain conditions

---

regret under this different definition with an analysis specifically tailored to our algorithm, which is deferred to Appendix B. Furthermore, since the transition kernel in the infinite-horizon setting is invariant over time and the optimal policy itself is also stationary, it is more natural to compare the optimal policy to a stationary policy, e.g., the policy $\pi_t$ deployed by the algorithm at each step, as in (3). Before this work, this has also been recently studied in [49; 18].

**Notation.** Given any vector $x \in \mathbb{R}^{SA}$ that represents a function $x : \mathcal{S} \times \mathcal{A} \to \mathbb{R}$, we use $x(s, a)$ to denote the entry corresponding to the state-action pair $(s, a)$. We also denote the probability transition vector at $(s, a)$ with

$$P_{s,a} = P(\cdot \mid s, a) \in \mathbb{R}^{1 \times S}, \tag{4}$$

that is, given any $V \in \mathbb{R}^S$, $P_{s,a}V = \mathbb{E}_{s' \sim P(\cdot|s,a)}[V(s')]$. For two vectors $x, y \in \mathbb{R}^{SA}$, we override the notation $x \le y$ to mean that $x(s, a) \le y(s, a)$ in every dimension $(s, a)$.

## 3 Algorithm

In this section, we present our algorithm Q-SlowSwitch-Adv and some relevant discussion.

---

**Algorithm 2:** Auxiliary functions

1 **Function** `update-q-ucb()`:

2 $\quad$ $Q^{\mathrm{UCB}}(s_t, a_t) \leftarrow (1 - \eta_n)Q^{\mathrm{UCB}}(s_t, a_t) + \eta_n\Big(r(s_t, a_t) + \gamma V(s_{t+1}) + c_b\sqrt{\frac{\iota}{(1-\gamma)^3 n}}\Big).$

3 **Function** `update-q-lcb()`:

4 $\quad$ $Q^{\mathrm{LCB}}(s_t, a_t) \leftarrow (1 - \eta_n)Q^{\mathrm{LCB}}(s_t, a_t) + \eta_n\Big(r(s_t, a_t) + \gamma V^{\mathrm{LCB}}(s_{t+1}) - c_b\sqrt{\frac{\iota}{(1-\gamma)^3 n}}\Big).$

5 **Function** `update-q-lazy()`:

6 $\quad$ **for** every $((s, a), q) \in \mathcal{D}$ **do** $Q^{\mathrm{lazy}}(s, a) \leftarrow q.$ $\quad$ # Update execution policy with the buffer

7 **Function** `update-q-reference()`:

8 $\quad$ $[\mu^{\mathrm{ref}}, \sigma^{\mathrm{ref}}](s_t, a_t) \leftarrow$ **update-moments**();

9 $\quad$ $[\delta^{\mathrm{R}}, B^{\mathrm{R}}](s_t, a_t) \leftarrow$ **update-reference-bonus**();

10 $\quad$ $b^{\mathrm{R}} \leftarrow B^{\mathrm{R}}(s_t, a_t) + (1 - \eta_n)\frac{\delta^{\mathrm{R}}(s_t, a_t)}{\eta_n} + c_b\frac{\iota^2}{n^{3/4}(1-\gamma)^2};$

11 $\quad$ $Q^{\mathrm{R}}(s_t, a_t) \leftarrow$
$\quad$ $(1 - \eta_n)Q^{\mathrm{R}}(s_t, a_t) + \eta_n\left(r(s_t, a_t) + \gamma\left(V(s_{t+1}) - V^{\mathrm{R}}(s_{t+1}) + \mu^{\mathrm{ref}}(s_t, a_t)\right) + b^{\mathrm{R}}\right).$

12 **Function** `update-moments()`:

13 $\quad$ $\mu^{\mathrm{ref}}(s_t, a_t) \leftarrow (1 - \frac{1}{n})\mu^{\mathrm{ref}}(s_t, a_t) + \frac{1}{n}V^{\mathrm{R}}(s_{t+1});$ $\quad$ # Running mean of the reference

14 $\quad$ $\sigma^{\mathrm{ref}}(s_t, a_t) \leftarrow (1 - \frac{1}{n})\sigma^{\mathrm{ref}}(s_t, a_t) + \frac{1}{n}\left(V^{\mathrm{R}}(s_{t+1})\right)^2;$ $\quad$ # Running 2nd moment of the reference

15 $\quad$ $\mu^{\mathrm{adv}}(s_t, a_t) \leftarrow (1 - \eta_n)\mu^{\mathrm{adv}}(s_t, a_t) + \eta_n\left(V(s_{t+1}) - V^{\mathrm{R}}(s_{t+1})\right);$ $\quad$ # Running mean of the advantage

16 $\quad$ $\sigma^{\mathrm{adv}}(s_t, a_t) \leftarrow (1 - \eta_n)\sigma^{\mathrm{adv}}(s_t, a_t) + \eta_n\left(V(s_{t+1}) - V^{\mathrm{R}}(s_{t+1})\right)^2.$ # Running 2nd moment of the advantage

17 **Function** `update-reference-bonus()`:

18 $\quad$ $B^{\mathrm{next}}(s_t, a_t) \leftarrow$
$\quad$ $c_b\sqrt{\frac{\iota}{n}}\left(\sqrt{\sigma^{\mathrm{ref}}(s_t, a_t) - \left(\mu^{\mathrm{ref}}(s_t, a_t)\right)^2} + \frac{1}{\sqrt{1-\gamma}}\sqrt{\sigma^{\mathrm{adv}}(s_t, a_t) - \left(\mu^{\mathrm{adv}}(s_t, a_t)\right)^2}\right);$

19 $\quad$ $\delta^{\mathrm{R}}(s_t, a_t) \leftarrow B^{\mathrm{next}}(s_t, a_t) - B^{\mathrm{R}}(s_t, a_t);$

20 $\quad$ $B^{\mathrm{R}}(s_t, a_t) \leftarrow B^{\mathrm{next}}(s_t, a_t).$

---

### 3.1 Review: Q-Learning with UCB and reference advantage

First, we make a brief review of the Q-learning with UCB method proposed in [15], referred to as UCB-Q hereafter, and its variance-reduced variant UCB-Q-Advantage, later introduced in [48]. The Q-function updates in Q-SlowSwitch-Adv are inspired by these two methods.

**Q-learning with UCB** The original Q-learning [41; 40] is a fixed-point iteration based on a stochastic approximation of the Bellman optimality equation (2). It uses a greedy policy with respect to its estimate of $Q^\star$, whose update rule can be summarized as:

$$Q(s, a) \leftarrow (1 - \eta)Q(s, a) + \eta\left(r(s, a) + \gamma\widehat{P}_{s,a}V\right). \tag{5}$$

Above, $Q$ (resp. $V$) is the running estimate of $Q^\star$ (resp. $V^\star$); $\eta \in (0, 1]$ is the (possibly varying) learning rate; $\widehat{P}_{s,a}V$ is a stochastic approximation of $P_{s,a}V$ (cf. (4)). Commonly, $V(s')$ is used for $\widehat{P}_{s,a}V$ in (5) as an unbiased estimate of $P_{s,a}V$, when a sample of state transition from $(s, a)$, namely $(s, a, s')$, is available.

However, as [15] point out, using (5) naïvely suffers from great regret suboptimality, for it rules out the state-action pairs with high value but few observations. To promote the exploration of such state-action pairs, UCB-Q appends (5) with an exploration bonus. Its update rule can be written as:

$$Q^{\mathrm{UCB}}(s, a) \leftarrow (1 - \eta)Q^{\mathrm{UCB}}(s, a) + \eta\left(r(s, a) + \gamma\widehat{P}_{s,a}V + b\right). \tag{6}$$

To encourage exploration, the bonus $b \geq 0$ is designed to maintain an upper confidence bound (UCB) on $(\widehat{P}_{s,a} - P_{s,a})V$, which in turn keeps $Q^{\mathrm{UCB}}(s, a)$ as an "optimistic" overestimate of $Q^\star(s, a)$.

**Q-learning with UCB and reference advantage** The regret guarantee for UCB-Q is still shy of being optimal. In order to attain optimality, one can turn to the celebrated idea of variance reduction [16; 23; 34; 37], which decomposes the stochastic approximation target into two parts: a low-variance reference estimated with batches of samples and a low-magnitude advantage estimated with every new sample. In this spirit, [48] introduce UCB-Q-Advantage based on UCB-Q and a reference-advantage decomposition. Specifically, given a reference $V^{\mathrm{R}}$ that is maintained as an approximation for $V^\star$, the update rule of UCB-Q-Advantage reads:

$$Q^{\mathrm{R}}(s,a) \leftarrow (1-\eta)Q^{\mathrm{R}}(s,a) + \eta\Big(r(s,a) + \gamma\big(\widehat{P}_{s,a}\big(V - V^{\mathrm{R}}\big) + \widehat{PV^{\mathrm{R}}}(s,a)\big) + b^{\mathrm{R}}\Big). \quad (7)$$

This idea of (7) is used in subroutine **update-q-reference**() of our algorithm. Let us discuss it in more details:

- Given a transition sample $(s,a,s')$, we can take $V(s') - V^{\mathrm{R}}(s')$ as an unbiased estimate of the advantage $P_{s,a}(V - V^{\mathrm{R}})$. The magnitude of $V - V^{\mathrm{R}}$ is small when $V$ and $V^{\mathrm{R}}$ are close. This engenders smaller stochastic volatility, compared to $\widehat{P}_{s,a}V$ in (6) from UCB-Q.

- The reference estimate $\widehat{PV^{\mathrm{R}}}$ is a stochastic approximation of $PV^{\mathrm{R}}$. In our algorithm, the auxiliary estimate $\mu^{\mathrm{ref}}$ (cf. Line 13 of Algorithm 2) is used as the estimate for $PV^{\mathrm{R}}$. Specifically, $\mu^{\mathrm{ref}}(s,a)$ is a running mean of $P_{s,a}V^{\mathrm{R}}$ based on the samples from all past visitations of $(s,a)$. In contrast to the advantage, which is computed every time a new sample arrives, the reference is computed with a batch of samples and thus more stable. In sacrifice, the reference is only updated intermittently and not as up-to-date as the advantage.

The exploration bonus $b^{\mathrm{R}}$ is computed from the auxiliary estimates in Line 8 and 9 to serve as an upper confidence bound on the aggregation of the aforementioned reference and advantage. Specifically, for each $(s,a)$, $\mu^{\mathrm{ref}}(s,a)$ and $\sigma^{\mathrm{ref}}(s,a)$ are the running mean and 2nd moment of the reference $[PV^{\mathrm{R}}](s,a)$ respectively; $\mu^{\mathrm{adv}}(s,a)$ and $\sigma^{\mathrm{adv}}(s,a)$ are the running mean and 2nd moment of the advantage $[P(V - V^{\mathrm{R}})](s,a)$ respectively; $B^{\mathrm{R}}(s,a)$ combines the empirical standard deviations of the reference and the advantage; $\delta^{\mathrm{R}}(s,a)$ is the temporal difference between $B^{\mathrm{R}}(s,a)$ and its previous value. $b^{\mathrm{R}}(s,a)$ can be computed from these estimates as a temporally-weighted average of $B^{\mathrm{R}}(s,a)$. Thanks to the low variability of the reference $PV^{\mathrm{R}}$, we can obtain a more accurate, milder overestimation in the upper confidence bound for faster overall convergence.

## 3.2 Review: Early settlement of reference value

Given the optimistic overestimates $Q^{\mathrm{UCB}}$ and $Q^{\mathrm{R}}$, it is natural to design an update rule of our Q-function estimate as the minimum of the estimate itself and these two overestimates (Line 9 of Algorithm 1). This makes our Q-function estimate monotonically decrease without violating the optimistic principle $Q \geq Q^\star$, which effectively enables us to lessen the overestimation in $Q$ over time until it converges to $Q^\star$. In fact, this is precisely the update rule in UCB-Q-Advantage [48]. Nevertheless, we need to equip our algorithm with additional features to strive for regret optimality.

[23] introduced a way to update the reference with higher sample efficiency in the finite-horizon setting. As they noted, it is critical to update the reference $V^{\mathrm{R}}$ in a smart fashion so as to balance the tradeoff between its synchronization with $V$ and the volatility that results from too many stochastic updates. Concretely, the reference $V^{\mathrm{R}}$ needs to be updated in a timely manner from $V$ so that the magnitude of $\widehat{P}_{s,a}(V - V^{\mathrm{R}})$ can be kept low as desired, but the updates cannot be too frequent either, because the stochasticity or variance in $\widehat{PV^{\mathrm{R}}}(s,a)$ could be as high as that in $\widehat{P}_{s,a}V$ of (6) and thus lead to suboptimality if it is not carefully controlled. To resolve this dilemma, we can update $V^{\mathrm{R}}$ until it becomes sufficiently close to $V^\star$ and fix its value thereafter.

To this end, we maintain a "pessimistic" underestimate $Q^{\mathrm{LCB}}$ (resp. $V^{\mathrm{LCB}}$) of $Q^\star$ (resp. $V^\star$) in the algorithm, which are computed from the lower confidence bound for $Q^\star$ (resp. $V^\star$). This can provide us with an upper bound on $V^{\mathrm{R}} - V^\star$, which will be used to determine when the update of the reference $V^{\mathrm{R}}$ should be stopped.

In particular, the if-else block in Line 23 to 26 is designed to keep the reference $V^{\mathrm{R}}$ synchronized with $V$ for each state $s$ respectively and terminate the update once

$$V(s) \leq V^{\mathrm{LCB}} + 3 \leq V^\star + 3. \quad (8)$$

This can guarantee $|V - V^{\mathrm{R}}| \leq 6$ throughout the execution of the algorithm. As a result, the standard deviation of $\widehat{P}_{s,a}(V - V^{\mathrm{R}})$ is guaranteed to be $O(1)$, which can be $O(\frac{1}{1-\gamma})$ times smaller than the standard deviation of $\widehat{P}_{s,a}V$ in (2). This can lead to smaller $\frac{1}{1-\gamma}$ factor in the final regret guarantee.

## 3.3 Adaptive low-switching greedy policy

Although these aforementioned designs from the finite-horizon literature help increase the accuracy of our estimate $Q$, they are still insufficient to attain regret optimality in the infinite-horizon setting. Since data collection takes place over a single trajectory with no reset, drastic changes in the execution policy can inflict a long-lasting volatility on the future trajectory and slow down the convergence. This is precisely the difficulty of the infinite-horizon setting over the finite-horizon one. The need to control the trajectory variability motivates us to design a novel adaptive switching technique.

Recall in UCB-Q and UCB-Q-Advantage, the execution policy is greedy with respect to the estimate $Q$, i.e., $\pi_t(s_t) = \arg\max_a Q(s_t, a)$. Every time $Q$ gets updated, what the algorithm effectively does is to make an estimate of $Q^{\pi_t}$ with the samples generated by $\pi_t$. Such $Q^{\pi_t}$ is only estimated and updated once before the execution policy is switched to $\pi_{t+1}$. This seems insufficient from a stochastic fixed-point iteration perspective, so we seek to update it more and learn each $Q^{\pi_t}$ better before switching to a new policy.

To tackle this issue, we make the execution policy $\pi_t$ greedy to $Q^{\mathrm{lazy}}$, which is updated lazily yet adaptively in Q-SlowSwitch-Adv. Specifically, for every $(s, a)$, we use $\theta(s, a)$ (cf. Line 12 in Algorithm 2) to keep track of the cumulative difference between the current $Q(s, a)$ and $Q_M(s, a)$, the latter of which is defined to be the value of $Q(s, a)$ last time $Q^{\mathrm{lazy}}$ is updated immediately after visiting $(s, a)$. Whenever $\theta(s, a)$ exceeds $\frac{1}{1-\gamma}$, indicating $Q^{\mathrm{lazy}}(s, a)$ and the execution policy has become outdated with respect to the current $Q(s, a)$, we reset $\theta(s, a)$ and set $u^{\mathrm{switch}}$ to True, which will direct the algorithm to update the entire function $Q^{\mathrm{lazy}}$ in the following step. **update-q-lazy()** updates $Q^{\mathrm{lazy}}$ with the samples from $\mathcal{D}$, which is a dictionary that serves as a buffer to store all the new sample transitions and their latest estimates since the last update of $Q^{\mathrm{lazy}}$.

In contrast, conventional low-switching algorithms update the execution policy on a predetermined, exponentially phased schedule [5; 48]. While trajectory stability is attained with these algorithms, as time goes on, it takes them exponentially longer to update policy, making them oblivious to recent large updates in the estimated Q-function. This would lead to suboptimal regret in the infinite-horizon setting, as continual choices of suboptimal actions will keep a lasting effect on future trajectory in the absence of trajectory reset. This issue is overcome in our algorithm by ignoring minor changes in function $Q$ yet still being adaptive to substantial changes in any state-action pair.

## 4  Main Results

Our model-free algorithm Q-SlowSwitch-Adv can achieve optimal regret with short burn-in time. Its theoretical guarantee can be summarized in the following theorem.

**Theorem 1.** *Choose any $\delta \in (0, 1)$. Suppose $\iota = \log \frac{SAT}{\delta}$ and $c_b$ is chosen to be a sufficiently large universal constant in Algorithm 1. Then with probability at least $1 - \delta$, Algorithm 1 achieves*

$$\mathrm{Regret}(T) \leq C_0 \left( \sqrt{\frac{SAT\iota^3}{(1-\gamma)^3}} + \frac{SA\iota^{\frac{7}{2}}}{(1-\gamma)^8} \right) \tag{9}$$

*for an absolute constant $C_0 > 0$.*

To prove this theorem, we need to use a recursive error decomposition scheme different from the existing work. The stationary nature of the infinite-horizon setting gives rise to several error terms unique to the infinite-horizon setting, and our novel switching technique is crucial at controlling them optimally. The proof is provided in the full version [14]. Now let us highlight a few key properties of our algorithm.

**Optimal regret with low burn-in.**  Q-SlowSwitch-Adv achieves optimal regret modulo some logarithmic factor as soon as the sample size $T$ exceeds

$$T \geq \frac{SA}{\text{poly}(1-\gamma)}.$$ (10)

This burn-in threshold is significantly lower than the $\frac{S^3 A^2}{\text{poly}(1-\gamma)}$ threshold in [12] when $SA \gg \frac{1}{1-\gamma}$. In other words, in the regime of (10), the regret of Q-SlowSwitch-Adv is guaranteed to satisfy

$$\text{Regret}(T) \leq \widetilde{O}\left(\sqrt{\frac{SAT}{(1-\gamma)^3}}\right),$$ (11)

which matches the lower bound in Table 1.

**Sample complexity.**  As a corollary of Theorem 1, it can be seen that Q-SlowSwitch-Adv attains $\epsilon$-average regret (i.e. $\frac{1}{T}\text{Regret}(T) \leq \epsilon$ for any fixed $T$) with sample complexity

$$\widetilde{O}\left(\frac{SA}{(1-\gamma)^3 \epsilon^2}\right).$$ (12)

This is lower than the sample complexity of the model-free algorithm in [24], which is $\widetilde{O}\left(\frac{SA}{(1-\gamma)^5 \epsilon^2}\right)$. Moreover, (12) holds true for any desired accuracy $\epsilon \in \left(0, \frac{(1-\gamma)^{13}}{SA}\right]$. This is a broader range than the ones in [12; 49], which involve higher order of $S$ and $A$ and only allow their algorithms to attain their respective optimal sample complexity in the high-precision regime.

**Space complexity.**  Q-SlowSwitch-Adv is a model-free algorithm that keeps a few estimates of the Q-function during execution, so its memory cost is as low as $O(SA)$. This is not improvable in the tabular setting, since it requires $O(SA)$ units of space to store the optimal policy. In contrast, the model-based UCBVI-$\gamma$ in [12] stores an estimate of the probability transition kernel and thus incurs a higher memory cost of $O(S^2 A)$.

**Computational complexity.**  The computational cost of Q-SlowSwitch-Adv is only $O(T)$. This is on the same order as reading samples along the $T$-length executed trajectory and is thus unimprovable. In comparison, our algorithm has a considerably lower computational cost than the one in [12], which requires $O(S^2 AT)$ operations overall.

## 5   Discussion

This work has introduced a model-free algorithm that achieves optimal regret in infinite-horizon discounted MDPs, which reduces the space and computational complexity requirement for regret optimality in the existing work. It also achieves optimal sample efficiency with a short burn-in time compared to other algorithms, including [12; 49]. Moreover, our algorithm has demonstrated the importance of switching policies slowly in infinite-horizon MDPs and introduced a novel technique might be of additional interest to future work. While our burn-in threshold is considerably reduced with respect to the order of $S$ and $A$, it still has nontrivial suboptimality in the effective horizon $\frac{1}{1-\gamma}$, which is a price we pay for using the reference-advantage technique. It is open for future work to investigate how to further improve the effective horizon factors in the burn-in cost.

## Acknowledgements

The authors are very grateful to Yuxin Chen for helpful advice as well as suggesting the direction. X. Ji was supported in part by the NSF grants 1907661 and 2014279.

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

# A    Related Work

In this section, we take a moment to discuss the related work beyond those in Table 1.

**Regret analysis for online episodic RL**    In the online episodic setting, regret is the predominant choice of metric for demonstrating the sample efficiency of a method [45; 28; 13]. [4] was the first to introduce a model-based method that can achieve near-optimal regret guarantee, but the model-based nature of their method induces a high space complexity and burn-in time. On the other hand, model-free methods are proposed in [15; 5], which are motivated by Q-learning and thus enjoy a low space complexity. However, these methods cannot guarantee optimal regret. It was not until [48] that proposed the first model-free method with optimal regret guarantee UCB-Q-Advantage, but it incurs a large burn-in time of $S^6 A^4 H^{28}$, where $H$ is the horizon of the episodic MDP. In addition, [26] proposed UCB-M-Q, a Q-learning variant with momentum, which can achieve optimal regret with low burn-in time, but it requires the storage of all momentum bias and thus incurs high memory cost. Recently, [23] propose a Q-learning variant with variance reduction that achieves optimal regret with $O(SAH)$ space complexity and $SA\mathrm{poly}(H)$ burn-in threshold at the same time. Table 1 in [23] provides a more detailed comparison of related work from the online episodic RL literature.

**Sample complexity for infinite-horizon RL**    In the infinite-horizon setting, there exist other sample efficiency metrics besides sample complexity of exploration. Initially, [19] considered the sample complexity needed to find an $\epsilon$-approximate optimal policy. The same definition is also considered in [38; 34; 33; 21]. Later, [37] studied the sample complexity needed to find an $\epsilon$-approximate optimal Q-function. Note that all of these works assume the generative setting. Indeed, a limitation of these aforementioned sample complexity definitions is that they only measure the performance of the final output policy and do not reflect the online regret during learning. Thus, existing works that study the online setting consider sample complexity of exploration and cumulative regret instead.

**Variance reduction in RL**    The idea of variance reduction was first introduced to accelerate stochastic finite-sum optimization by [16], which is followed by a rich literature [42; 27; 11]. Later, for better sample efficiency in RL, it is applied to policy gradient methods [25; 47; 29] as well as value-based methods in various problems including generative setting RL [33; 34; 37], policy evaluation [9; 43], asynchronous Q-learning [22; 44] and offline RL [46; 32].

**Low-switching algorithms**    Since our algorithm includes a novel feature that switches the execution policy slowly, we make a review of the low-switching approaches in RL. The idea of changing the execution policy infrequently during learning was first introduced by [3] as an approach to minimize regret in the multi-armed bandit problem. [5] adapted this idea to tabular RL and formalized the switching cost as a secondary metric that an algorithm can minimize. To reduce the number of policy switches and thus the switching cost, their algorithm updates the policy in geometrically longer intervals. Similar techniques can be found in [48], whose algorithm can achieve regret optimality while maintaining low switching cost. Later, [10; 39] introduced a new low-switching approach in linear MDPs by switching policies when the estimated covariance matrix gets a significant update. All these methods guarantee a $O(\log T)$ switching cost. The switching cost guarantee was later improved to $O(\log \log T)$ by the algorithms proposed in [31; 50].

# B    A Discussion about Different Regret Metrics

As we have discussed earlier, multiple metrics exist in the existing literature for evaluating the online performance of an RL algorithm in the infinite-horizon setting. In fact, [12] and [24] both use a respectively different regret metric from the one in this work and [18]. An argument showing the equivalence between the regret metric in [24] and the one in [12] is discussed in Appendix A.2 of [12]. For this reason and the fact that the regret guarantee in [24] is not minimax-optimal, we focus on the relation between our theoretical guarantee in Theorem 1 and the regret metric from [12] in this section.

Recall that the goal of RL is to learn the optimal policy through online interactions with the environment (and there always exists a stationary optimal policy), so the regret metric we consider in this

work is

$$\text{Regret}(T) := \sum_{t=1}^{T} \left( V^{\star}(s_t) - V^{\pi_t}(s_t) \right), \tag{13}$$

where $\pi_t$ is the stationary policy that the algorithm uses to take action $a_t$ at step $t$.

The non-stationary regret metric considered in [12] is

$$\text{Regret}_{\text{NS}}(T) := \sum_{t=1}^{T} \left( V^{\star}(s_t) - V^{\{\pi_j\}_{j=t}^{\infty}}(s_t) \right). \tag{14}$$

Here, $V^{\{\pi_j\}_{j=t}^{\infty}}(s) := \mathbb{E}\left[ \sum_{i=0}^{\infty} \gamma^i r(s_i, \pi_{t+i}(s_i)) \mid s_0 = s \right]$, which is the expected cumulative future reward of the non-stationary algorithm starting from time $t$.

In fact, similar difference can also be observed in the literature focused on sample complexity of exploration. Notably, the metric in [49] compares the optimal value function $V^{\star}(s_t)$ against $V^{\pi_t}(s_t)$, while the metric in works such as [35; 8] compares against $V^{\{\pi_j\}_{j=t}^{\infty}}(s_t)$.

There is no formal equivalence between the two regret metrics in general, despite the intuition we have provided in Remark 1 that $\text{Regret}_{\text{NS}}(T)$ should be smaller than $\text{Regret}(T)$ for any improving algorithm. However, we can show that the specific algorithm Q-SlowSwitch-Adv (Algorithm 1) achieves $\widetilde{O}(\sqrt{\frac{SAT}{(1-\gamma)^3}})$ regret under the regret metric defined in [12], which matches the lower bound in [12] and is thus optimal for this metric as well.

Before the analysis, let us first define a notation. Let $f$ and $g$ be two real-valued functions that take $\mathcal{X} := (S, A, \gamma, T, \frac{1}{\delta})$ as arguments. If there exists a universal constant $C > 0$ such that $f(\mathcal{X}) \leq Cg(\mathcal{X})$ for any instantiation of $\mathcal{X}$, we can denote this with the notation $f(\mathcal{X}) \lesssim g(\mathcal{X})$.

Let us define

$$\mathcal{T} = \{1 \leq t \leq T : \pi_t \neq \pi_{t+1}\}, \tag{15}$$

which is the set of time indices that the execution policy switches in the following step, and

$$\mathcal{T}_H = \{1 \leq t \leq T : t + h \in \mathcal{T} \text{ for some } 0 \leq h \leq H\}, \tag{16}$$

which is the set of time indices that the execution policy switches in any of the following $H$ steps.

First, notice

$$\left| \sum_{t \notin \mathcal{T}_H} \left( V^{\{\pi_j\}_{j=t}^{\infty}}(s_t) - \sum_{i=0}^{H} \gamma^i r(s_{t+i}, a_{t+i}) \right) \right|$$

$$\leq \left| \sum_{t \notin \mathcal{T}_H} \left( \mathbb{E}\left[ \sum_{i=0}^{H} \gamma^i r(s_{t+i}, a_{t+i}) \right] - \sum_{i=0}^{H} \gamma^i r(s_{t+i}, a_{t+i}) \right) \right| + \frac{\gamma^H T}{1-\gamma}$$

$$\leq \sum_{k=1}^{H} \left| \sum_{t = jH+k \notin \mathcal{T}_H} \left( \mathbb{E}\left[ \sum_{i=0}^{H} \gamma^i r(s_{t+i}, a_{t+i}) \right] - \sum_{i=0}^{H} \gamma^i r(s_{t+i}, a_{t+i}) \right) \right| + \frac{1}{T}$$

$$\lesssim \sqrt{\frac{T \log^3 T}{(1-\gamma)^3}}, \tag{17}$$

where the second line holds when $H \gtrsim \frac{\log T}{1-\gamma}$, and the last line makes use of the Azuma-Hoeffding inequality.

Similarly, we have

$$\left| \sum_{t \notin \mathcal{T}_H} \left( V^{\pi_t}(s_t) - \sum_{i=0}^{H} \gamma^i r(s_{t+i}, a_{t+i}) \right) \right| \lesssim \sqrt{\frac{T \log^3 T}{(1-\gamma)^3}} \tag{18}$$

since $\pi_t = \pi_{t+i}$ for $0 \leq i \leq H$ and $t \notin \mathcal{T}_H$.

Putting (17) and (18) together leads to

$$\left| \text{Regret}(T) - \text{Regret}_{\text{NS}}(T) \right| = \left| \sum_{t=1}^{T} \left( V_t^{\pi_t}(s_t) - V^{\{\pi_j\}_{j=t}^{\infty}}(s_t) \right) \right|$$

$$\lesssim \sqrt{\frac{T \log^3 T}{(1-\gamma)^3}} + \frac{|\mathcal{T}| \log T}{(1-\gamma)^2} \tag{19}$$

by noticing that $\left| \sum_{t \in \mathcal{T}_H} \left( V^{\pi_t}(s_t) - V^{\{\pi_j\}_{j=t}^{\infty}}(s_t) \right) \right| \lesssim \frac{|\mathcal{T}_H|}{1-\gamma} \lesssim \frac{|\mathcal{T}| \log T}{(1-\gamma)^2}$.

Since our algorithm is low-switching and $|\mathcal{T}| \leq \widetilde{O}(\frac{SA}{(1-\gamma)^{9/2}} + \frac{(SA)^{3/4} T^{1/4}}{(1-\gamma)^{5/4}})$, the difference between the two regret metrics (19) is dominated by the upper bound on the regret itself. Thus, given our result from Theorem 1 that $\text{Regret}(T) \leq \widetilde{O}(\sqrt{\frac{SAT}{(1-\gamma)^3}})$, we can conclude $\text{Regret}_{\text{NS}}(T) \leq \widetilde{O}(\sqrt{\frac{SAT}{(1-\gamma)^3}})$ for our algorithm Q-SlowSwitch-Adv. In fact, this conversion holds as long as the algorithm's switching cost is dominated by the regret itself, e.g., when the switching cost is $o(\sqrt{T})$.

