# OpenReview forum: "Regret-Optimal Model-Free Reinforcement Learning for Discounted MDPs with Short Burn-In Time"
_NeurIPS.cc/2023/Conference — NeurIPS 2023 poster_

### Official Review · Reviewer_pMQU · 2023-06-24

**Soundness:** 4 excellent
**Presentation:** 4 excellent
**Contribution:** 3 good
**Rating:** 7
**Confidence:** 4

**Summary:**

The paper proposes a nearly optimal model-free algorithm in discounted tabular MDPs which has a shorter burn-in time than previous works and provides valid theoretical guarantees.


**Strengths:**

The paper is well motivated, trying to shorten the burn-in time and obtain a more sample-efficient model-free tabular MDP algorithm.

The writing is awesome. The detailed comparison with previous algorithms is highly appreciated.

The proof seems correct with many previous techniques organically integrated together. I must say the author should be an expert on the related techniques and proof arguments.

To achieve a tighter burn-in time, it seems vital to update the reference advantage at an appropriate frequency and learn the Q-value of the current policy more accurately. To tackle the issue, the author proposes two new components. The first is the lazy update for the Q-value function which the greedy policy is generated; the other is an adaptive low-switching mechanism to determine what is the time to perform such an update. Both components might inspire future works.

**Weaknesses:**

If I must point out some weaknesses, I think that the proof contains too many instances where the author didn’t include brackets when utilizing the summation symbol $\sum$, such as in (46)-(48).

The frequency of these omissions is quite high, making it difficult to enumerate them here.  I think it is important that all brackets are appropriately included for clarity and correctness.

I have some questions about the results and the theoretical analysis.

**Questions:**

1. The author discusses the impact of large state-action spaces on burn-in time several times. It is reasonable to expect that tabular MDPs with large state-action spaces are generally more challenging to learn, leading to longer burn-in times. However, the paper lacks a lower bound for the minimum burn-in time, although it does match the well-known lower bound on the dominant term (or variance term). It would be valuable for the author to address this gap and provide insights into the possible smallest burn-in time.

2. Lemma 10 presents a rough estimate for $\sum_{t=1}^T (V_{t-1}\left(s_t\right)-V^{\pi_t}\left(s_t\right))$, which is subsequently utilized multiple times in the proof of Lemma 7 (e.g., in (153), (228), (229)). Instead of using this crude bound repeatedly, it seems to be more advantageous to bound the regret without this rough estimate. I mean to maintain the dependence of $\sum_{t=1}^T (V_{t-1}\left(s_t\right)-V^{\pi_t}\left(s_t\right))$ while analyzing intermediate terms in Lemma 4 and then to use Proposition 1 to disentangle this dependence at the end. By using this alternative argument, it may be possible to eliminate the need for the crude bound entirely. Additionally, it is worth investigating if this alternative argument can enhance the dependence of $(1-\gamma)^{-1}$ on the burn-in time.

3. It is notable that the regret of UCBVI-$\gamma$ exhibits a more favorable dependence on $(1-\gamma)^{-1}$ compared to the algorithm proposed in this work. It would greatly enhance the paper if the author could provide a discussion on the reasons behind this superiority. Understanding the factors contributing to better dependence would provide valuable insights into the comparative performance of the two algorithms and contribute to a deeper understanding of their respective strengths and weaknesses.






**Limitations:**

Please see the questinos.

---

> ### Author Rebuttal · Authors · 2023-08-10
>
> We really appreciate you taking the time to provide this valuable review! Let us provide responses to your comments and questions as follows.
>
> > If I must point out some weaknesses, I think that the proof contains too many instances where the author didn’t include brackets when utilizing the summation symbol $\sum$, such as in (46)-(48).
> The frequency of these omissions is quite high, making it difficult to enumerate them here. I think it is important that all brackets are appropriately included for clarity and correctness.
>
> Thank you for the suggestion! We will correct these in the final version of our paper.
>
> > The author discusses the impact of large state-action spaces on burn-in time several times. It is reasonable to expect that tabular MDPs with large state-action spaces are generally more challenging to learn, leading to longer burn-in times. However, the paper lacks a lower bound for the minimum burn-in time, although it does match the well-known lower bound on the dominant term (or variance term). It would be valuable for the author to address this gap and provide insights into the possible smallest burn-in time.
>
> This is a good question. We had some discussion about what the lower bound suggests about burn-in cost in our introduction (towards the end of Section 1.3). In short, the current lower bound does not rule out the possibility of having a burn-in term as small as $\frac{SA}{(1-\gamma)^2}$. However, it is possible that this lower bound is not tight, so this can be a good open question for future work. We will add more discussion about this gap and the reason for our suboptimal $(1-\gamma)^{-1}$ dependence in the burn-in when we revise the paper.
>
> > Lemma 10 presents a rough estimate for $\sum_{t=1}^T (V_{t-1}(s_{t}) - V^{\pi_t}(s_t))$, which is subsequently utilized multiple times in the proof of Lemma 7 (e.g., in (153), (228), (229)). Instead of using this crude bound repeatedly, it seems to be more advantageous to bound the regret without this rough estimate. I mean to maintain the dependence of $\sum_{t=1}^T (V_{t-1}(s_{t}) - V^{\pi_t}(s_t))$ while analyzing intermediate terms in Lemma 4 and then to use Proposition 1 to disentangle this dependence at the end. By using this alternative argument, it may be possible to eliminate the need for the crude bound entirely. Additionally, it is worth investigating if this alternative argument can enhance the dependence of $(1-\gamma)^{-1}$ on the burn-in time.
>
> This is a very good observation. To be precise, (153) is the proof of Lemma 10 and not an invocation of the lemma, so we can focus on the other instances. We use Lemma 10 at these places because we want to replace $\sum_{t=1}^T (V_{t-1}(s_{t}) - V^{\pi_t}(s_t))$ with a simple upper bound without having to carry it around in the following analysis. Even if we carried $\sum_{t=1}^T (V_{t-1}(s_{t}) - V^{\pi_t}(s_t))$ all the way to the end without invoking Lemma 10 as you suggested, the burn-in cost would still have the current $(1-\gamma)^{-1}$ dependence. The invocations of Lemma 10 are only for clarity.
>
> Take (228) for a concrete example. (228) is an intermediate step to obtain the inequality in (216). Despite the use of Lemma 10, the $\log T$ term in (216) is only on the order of $(1-\gamma)^{-3}$. This is dominated by the $\log T$ term in (217), which is on the order of $(1-\gamma)^{-13/4}$, and this term in (217) is not a result of Lemma 10. Note that both (216) and (217) are part of $\mathcal{R}_1$, so using Lemma 10 or not does not affect the burn-in component of $\mathcal{R}_1$ or the final burn-in term.
>
> > It is notable that the regret of UCBVI-$\gamma$ exhibits a more favorable dependence on $(1-\gamma)^{-1}$ compared to the algorithm proposed in this work. It would greatly enhance the paper if the author could provide a discussion on the reasons behind this superiority. Understanding the factors contributing to better dependence would provide valuable insights into the comparative performance of the two algorithms and contribute to a deeper understanding of their respective strengths and weaknesses.
>
> This is also a good question. The reason why UCBVI-$\gamma$ has a lower $(1-\gamma)^{-1}$ dependence in the burn-in than ours is not so much what they did in their algorithm as what we did in ours that gives rise to a high $(1-\gamma)^{-1}$ dependence. To be specific, the high $(1-\gamma)^{-1}$ dependence comes from our use of the reference-advantage technique. In fact, similarly large factors of $1/(1−\gamma)$ can also be observed in the burn-in terms of other reference-advantage methods [1,2,3]. Mathematically, this can be attributed to the use of Cauchy-Schwarz in the analysis to balance the factors of $1/(1−\gamma)$ and $T$. For example, we need to manipulate a term like $T^{1/4}/(1−\gamma)^{13/4}$ into $\sqrt{T/(1−\gamma)^{3}} + 1/(1−\gamma)^{6}$ so that the $T$-dependent term can have the optimal $1/(1−\gamma)$ dependence at the expense of a larger burn-in term. Such uses of Cauchy-Schwarz can quickly amplify the power of $1/(1−\gamma)$ in the burn-in term. Note that terms like $T^{1/4}/(1−\gamma)^{13/4}$ are a result of the reference-advantage technique (e.g., higher-order variance terms like $(P-\widehat{P}) (V - V^{\mathrm{R}})$).
>
> [1] Li, G., Shi, L., Chen, Y., Gu, Y., and Chi, Y. Breaking the sample complexity barrier to regret-optimal model-free reinforcement learning. Advances in Neural Information Processing Systems, 34:17762–17776, 2021.
>
> [2] Zhang, Z., Zhou, Y., and Ji, X. Almost optimal model-free reinforcement learning via reference-advantage decomposition.
> Advances in Neural Information Processing Systems, 33:15198–15207, 2020.
>
> [3] Zhang, Z., Zhou, Y., and Ji, X. Model-free reinforcement learning: from clipped pseudo-regret to sample complexity. In International Conference on Machine Learning, pp. 12653–12662. PMLR, 2021.
>
> Please let us know if you have any further questions and we will be happy to answer. Thanks again!

---

> > ### Comment · Reviewer_pMQU · 2023-08-18
> >
> > Thanks for the response which addresses my questions well so I decide to keep my score.

---

### Official Review · Reviewer_Th7P · 2023-07-05

**Soundness:** 3 good
**Presentation:** 3 good
**Contribution:** 3 good
**Rating:** 6
**Confidence:** 3

**Summary:**

This paper studied infinite discounted tabular Markov decision process (MDP). The authors proposed a model-free algorithm: Q-SlowSwitch-Adv that achieves optimal regret with low burn-in cost, low space complexity and low computational cost. They introduce an innovative technique that changes the execution policy in a slow, adaptive manner, and employ variance reduction to enhance efficiency.

**Strengths:**

1. The major novelty of this work lies in the solution to achieve optimal regret using model-free algorithm for discounted tabular MDP.
2. The property of the low burn-in cost seems interesting and I believe that the low computation cost and low space complexity could be good contributions, enhancing the feasibility of implementing reinforcement learning in real-world scenarios.
3. The design of a slow, adaptive execution policy-switching technique is novel.


**Weaknesses:**

1. The paper lacks empirical evidence to substantiate the theoretical claims and the efficacy of the proposed algorithm, especially for the 'low' burn-in cost.
2. The reason why the proposed algorithm can reduce the burn-in cost is not clear, please see the Question part for more details.


**Questions:**

Could the authors elaborate on the methodology used to reduce the burn-in cost? Specifically, is this reduction achieved through refined theoretical analysis, novel algorithm design, or a combination of both?

**Limitations:**

1. The paper's main limitation is the lack of empirical evidence to support the proposed method. Performance comparisons against classic algorithms would have made the theoretical claims more credible.

2. Given the high level of complexity involved in real-world RL problems, the generalizability of the proposed method is not addressed. Therefore, it is unclear how the proposed algorithm will perform in more complex scenarios such as function approximation.

---

> ### Author Rebuttal · Authors · 2023-08-10
>
> We really appreciate you taking the time to provide this valuable review! Let us provide responses to your comments and questions as follows.
>
> > The paper lacks empirical evidence to substantiate the theoretical claims and the efficacy of the proposed algorithm, especially for the 'low' burn-in cost.
>
> Thank you for raising the point about empirical experiments. Our original goal is to resolve a longstanding theoretical optimality gap, but we agree that how an algorithm performs empirically can also be important. We will try to find an RL environment that can best demonstrate our point and conduct some experiments for the final version of our paper, since the rebuttal period is short.
>
> > Could the authors elaborate on the methodology used to reduce the burn-in cost? Specifically, is this reduction achieved through refined theoretical analysis, novel algorithm design, or a combination of both?
>
> This is a good question. This reduction can be mostly attributed to the algorithm. The model-free nature of our algorithm is a factor of this reduction. Since we only estimate the Q-function, which only has $SA$ entries, it is natural to see we can start making meaningful estimations as soon as samples exceed $\frac{SA}{\mathrm{poly}(1-\gamma)}$. The variance reduction technique and the novel adaptive slow-switching technique we used are also important. These techniques as well as their tight analysis ensure the dominant term in the regret is optimal without introducing extra $SA$ factors in the burn-in term.
>
> > Given the high level of complexity involved in real-world RL problems, the generalizability of the proposed method is not addressed. Therefore, it is unclear how the proposed algorithm will perform in more complex scenarios such as function approximation.
>
> Our result can provide some insights into more complex settings, such as how regret should scale with respect to each problem parameter and the fact that we need to control variance and policy switching carefully for optimality. While the general idea of our adaptive slow-switching technique is applicable to general function approximation, it is not obvious how it should be implemented exactly for that setting (the variance reduction idea we used has existed for a long time but does not have a general function approximation implementation yet). This is beyond the scope of this work and is a good question for future work to answer.
>
> Please let us know if you have any further questions and we will be happy to answer. Thanks again!

---

> > ### Comment · Reviewer_Th7P · 2023-08-15
> >
> > Thanks for the response. My concerns are addressed. I decide to keep my score.

---

### Official Review · Reviewer_an4o · 2023-07-06

**Soundness:** 4 excellent
**Presentation:** 3 good
**Contribution:** 4 excellent
**Rating:** 8
**Confidence:** 4

**Summary:**

This paper studies discounted infinite-horizon MDPs. The authors propose an online algorithm that attains optimal (i.e., least) regret with finite-sample performance guarantees.

**Strengths:**

Optimal regret & finite-sample performance guarantees

**Weaknesses:**

Lack of numerical illustration, but not a deal-break.

**Questions:**

1. Some numerical experiments in the main body would be great.
2. Assumptions on the probability distribution for the algorithm to achieve the optimal regret and finite-sample performance guarantees should be made explicitly and crystal clear.

**Limitations:**

Lack of numerical illustration, but not a deal-break.

---

> ### Author Rebuttal · Authors · 2023-08-10
>
> We really thank you for taking the time to provide this valuable review as well as your approval of our paper! Let us provide responses to your questions as follows.
>
> Thank you for raising the point about numerical experiments and illustration. Our original goal is to resolve a longstanding theoretical optimality gap, but we agree that how an algorithm performs empirically can also be important. We will try to find an RL environment that can best demonstrate our point and conduct some experiments for the final version of our paper, since the rebuttal period is short.
>
> > Assumptions on the probability distribution for the algorithm to achieve the optimal regret and finite-sample performance guarantees should be made explicitly and crystal clear.
>
> Thank you for raising this question. We did not make any assumption on the probability distribution. With a number $\delta$ decided, if we run the algorithm for $T$ steps (with no assumption), then Equation (9) in Theorem 1 holds with probability at least $1-\delta$. The transitions in the online trajectory is the only source of randomness in this problem, and no assumption is made about them.
>
> Please let us know if you have any further questions. Thanks again!

---

> > ### Comment · Reviewer_an4o · 2023-08-16
> >
> > I would like to thank the authors for their response. My concerns are well addressed.

---

> > > ### Author Response · Authors · 2023-08-16
> > >
> > > Thank you for your reply! We are happy that our response was able to help.

---

### Official Review · Reviewer_wb6F · 2023-07-10

**Soundness:** 3 good
**Presentation:** 3 good
**Contribution:** 3 good
**Rating:** 6
**Confidence:** 3

**Summary:**

The paper provides a version of Q-learning algorithm with Bernstein bonuses and control variate that allows or the correct (in terms of the dependence upon state and action space sizes $S$ and $A$) second-order term in the regret bound. This allows for an algorithm with the provably shorter burn-in period (again in terms of $S$ and $A$). This result for discounted MDPs is, to the best of my knowledge, novel.

**Strengths:**

The paper presents a valuable contribution to the theory of model-free algorithms for solving MDPs with finite state- action spaces. The suggested idea of incorporating Berstein style bonuses with the UCB-advantage type of control variates is appealing and novel in the discounted MDP setting.

**Weaknesses:**

Actually, I would also like to ask the authors why not considering submitting to a journal. The proof appendix is massive and it is obviously not possible to check all the technical details properly due to a very limited time for reviewing period.

Minor issues and comments:
1. The example concerning the Go game (lines 94-98) is spectacular, yet with $|S| = 3^{161}$ the linear dependence on $S$ is clearly as bad as $S^2$. I understand the author's willingness to explain the motivation beyond improving the dependence of the second-order term upon $S$ and $A#, yet I suggest to come up with a slightly less ambitious example;
2. I would also suggest removing "outstanding" in line 108: it is clearly a matter of taste;

**Questions:**

I suggest the authors to add explicitly a paragraph when discussing the techniqus currently used to obtain the second-order term in the regret whch is optimal either in terms of episode length $H$ (resp. $1-\gamma$ factor), or $S$ and $A$ factors. I would also suggest to add explicit comments on why the current technique in the paper is not suitable to get the second-order term, which is optimal also in terms of the $1-\gamma$ factor. With this discussion added, I can increase my score by $1$ point.

**Limitations:**

The paper is theoretical and I do not expect any negative societal impact of the paper.

---

> ### Author Rebuttal · Authors · 2023-08-10
>
> We really appreciate you taking the time to provide this valuable review! We also thank the reviewer for suggesting submission to a journal. We submitted it to this conference because we believed our paper can be seen by more people here and thus maximize its impact and contribution to the community. In addition, NeurIPS does not seem to discourage submissions with long proofs. But ultimately, this discussion is really beyond the scope of what we should discuss during rebuttal.
>
> > The example concerning the Go game (lines 94-98) is spectacular, yet with $|S| = 3^{161}$ the linear dependence on $S$ is clearly as bad as $S^2$. I understand the author's willingness to explain the motivation beyond improving the dependence of the second-order term upon $S$ and $A$, yet I suggest to come up with a slightly less ambitious example;
>
> This is a good point. Instead of Go, we can also take tic-tac-toe as an example. The state space for a 5x5 game is $3^{25}$. While it is feasible to store the Q-function in memory, anything on the order of $|\mathcal{S}|^2$ can be difficult. We will include this example in the final version of our paper when explaining the importance of finding a model-free approach.
>
> > I would also suggest removing "outstanding" in line 108: it is clearly a matter of taste;
>
> Thank you for the suggestion! We will remove that in the final version as you suggested.
>
> > I suggest the authors to add explicitly a paragraph when discussing the techniques currently used to obtain the second-order term in the regret which is optimal either in terms of episode length $H$ (resp. $1-\gamma$ factor), or $S$ and $A$ factors. I would also suggest to add explicit comments on why the current technique in the paper is not suitable to get the second-order term, which is optimal also in terms of the $1-\gamma$ factor. With this discussion added, I can increase my score by 1 point.
>
> We really appreciate your willingness to raise the score. It is certainly a good question. We weren’t able to discuss much about it in our initial submission due to the page limit, but we can certainly add this explanation in the final version, which will allow for an additional page.
>
> Let us first explain why our algorithm has a burn-in with high $\frac{1}{1-\gamma}$ dependence. This comes from our use of the reference-advantage technique. In fact, similarly large factors of $1/(1−\gamma)$ can also be observed in the burn-in terms of other reference-advantage methods [1,2,3]. Mathematically, this can be attributed to the use of Cauchy-Schwarz in the analysis to balance the factors of $1/(1−\gamma)$ and $T$. For example, we need to manipulate a term like $T^{1/4}/(1−\gamma)^{13/4}$ into $\sqrt{T/(1−\gamma)^{3}} + 1/(1−\gamma)^{6}$ so that the $T$-dependent term can have the optimal $1/(1−\gamma)$ dependence at the expense of a larger burn-in term. Such uses of Cauchy-Schwarz can quickly amplify the power of $1/(1−\gamma)$ in the burn-in term. Note that terms like $T^{1/4}/(1−\gamma)^{13/4}$ are a result of the reference-advantage technique (e.g., higher-order variance terms like $(P-\widehat{P}) (V - V^{\mathrm{R}})$). As to obtaining a burn-in with optimal dependence on $1/(1−\gamma)$, it is not clear if any existing algorithm can achieve this yet, not even UCBVI-$\gamma$ in [4]. It is a good future direction to study how to lower the $1/(1−\gamma)$ factors in the burn-in term when using reference-advantage algorithms.
>
> As to how we are able to achieve a linear dependence with $SA$ in the burn-in cost, the main reason is the model-free nature of our algorithm. Since we only estimate the Q-function, which only has $SA$ entries, it is natural to see we can start making meaningful estimations as soon as samples exceed $\frac{SA}{\mathrm{poly}(1-\gamma)}$. The variance reduction and adaptive slow-switching techniques we used are also important, as they ensure the dominant term in the regret is optimal without introducing extra $SA$ factors in the burn-in term.
>
> [1] Li, G., Shi, L., Chen, Y., Gu, Y., and Chi, Y. Breaking the sample complexity barrier to regret-optimal model-free reinforcement learning. Advances in Neural Information Processing Systems, 34:17762–17776, 2021.
>
> [2] Zhang, Z., Zhou, Y., and Ji, X. Almost optimal model-free reinforcement learning via reference-advantage decomposition.
> Advances in Neural Information Processing Systems, 33:15198–15207, 2020.
>
> [3] Zhang, Z., Zhou, Y., and Ji, X. Model-free reinforcement learning: from clipped pseudo-regret to sample complexity. In International Conference on Machine Learning, pp. 12653–12662. PMLR, 2021.
>
> [4] Jiafan He, Dongruo Zhou, and Quanquan Gu. Nearly minimax optimal reinforcement learning for discounted mdps. Advances in Neural Information Processing Systems, 34:22288–22300, 2021.
>
> In the very end, we just want to gently point out that getting an $SA$ dependence in the burn-in or second-order term is only one of our contributions. The other, perhaps more important, contribution is that our algorithm is the first model-free algorithm that achieves minimax-optimal regret $\widetilde{O}(\sqrt{\frac{SAT}{(1-\gamma)^3}})$. Compared to the state-of-the-art result in [4], we reduced the space complexity from $O(S^2A)$ to $O(SA)$ and the computational complexity from $O(S^2AT)$ to $O(T)$.
>
> Please let us know if you have any further questions and we will be happy to answer. Thanks again!

---

> > ### Author Response · Authors · 2023-08-18
> >
> > Dear Reviewer wb6F,
> >
> > Thank you again for taking the time to review our paper! You asked some very important questions in your review that we really hope to resolve. Could you please let us know if our rebuttal has addressed them sufficiently? If it has, could you kindly consider raising your score as you mentioned? Please don’t hesitate to contact us if you have any further questions.

---

> > ### Comment · Reviewer_wb6F · 2023-08-19
> >
> > Thank you for your detailed answer. I would raise my score to 6.

---

> > > ### Author Response · Authors · 2023-08-20
> > >
> > > Dear Reviewer wb6F,
> > >
> > > We are very grateful for your support. Thank you!

---

### Official Review · Reviewer_zms9 · 2023-07-20

**Soundness:** 3 good
**Presentation:** 2 fair
**Contribution:** 2 fair
**Rating:** 6
**Confidence:** 4

**Summary:**

The paper studies online learning in tabular infinite horizon MDPs, and proposes a model free algorithm that obtains (nearly) minimax optimal regret with space complexity $SA$, and such that the number of time steps needed to reach the minimax-optimality regime is $O(SA/(1-\gamma)^{13})$.
Prior art achieving minimax optimal regret required space complexity $S^2 A$ and $O(S^3A^2/(1-\gamma)^{4})$ time steps to reach the optimal regime.
It should be noted the definition of regret used here is not the same as that of prior art.

**Strengths:**

* The paper proposes a new approach to learning infinite horizon MDPs.
* The paper (perhaps) advances state of the art in online learning infinite horizon tabular MDPs.

**Weaknesses:**

- It is hard to say this work directly improves [14] because the objective considered is not the same. It would be helpful if a more formal discussion of this point was provided by the authors. For the same reason it is also unclear if the lower bound (also from [14]) applies. Given, this, the way the results are presented in the table are misleading.
- The improvement is rather incremental, shaving off a factor of $O(SA)$ (at the cost of an extra $(1-\gamma)^4$) from the lower order (non $T$ dependent) term. In addition, the algorithm is complicated and presentation does not make it particularly easy to understand.

**Questions:**

- Is there a more formal treatment of the two regret definitions? It would be helpful to make a more educated decision here; the improvement would have been marginal even if the objectives considered were the same, and when taking into account it may not even be an actual improvement --- the contribution seems questionable.
- It seems like the approach given here borrowed most of the algorithm from the finite horizon case [54] and then employs a slow switching strategy to essentially reduce the infinite horizon problem to a finite horizon one, thereby gaining in the $SA$ factors but loosing in the horizon. This also seems to be the source of the different regret definition. The presentation does not make this entirely clear however, is this more or less the case?

---

> ### Author Rebuttal · Authors · 2023-08-10
>
> We really appreciate you taking the time to provide this valuable review! Let us provide responses to your comments and questions as follows.
>
> > It is hard to say this work directly improves [14] ...
>
> > Is there a more formal treatment of the two regret definitions? It would be helpful to make a more educated decision here.
>
> This is a good question. While there is no discussion about this difference in the existing literature, let us provide some more explanation of the relationship between the two regret definitions here, in addition to our brief discussion in Section 2.3. The difference lies in the quantity being compared against the optimal value function at every step $t$. In the definition we used (also in [55,19]), (i) it is the expected cumulative reward of the execution policy $\pi_t$ at time $t$, i.e., $V^{\pi_t}(s) := E[\sum_{i=0}^\infty \gamma^i r(s_i, \pi_{t}(s_i)) | s_0=s]$. Note that $\pi_t$ is a stationary policy. In the other definition (of [14]), (ii) it is the expected cumulative future reward of the algorithm since time $t$. This can be viewed as the value function of the non-stationary policy $\\{\pi_j\\}\_{j=t}^\infty$, i.e., $V^{\\{\pi_j\\}\_{j=t}^\infty}(s) := E[\sum_{i=0}^\infty \gamma^i r(s_i, \pi_{t+i}(s_i)) | s_0=s]$. When the execution policy $\pi_t$ is becoming more and more optimal over the course of execution, which is expected given the theoretical results from [14] and our paper, it is clear that $V^{\\{\pi_j\\}\_{j=t}^\infty}$ of (ii) should be larger than $V^{\pi_t}$ of (i). This makes the regret $V^{\star}(s_t) - V^{\\{\pi_j\\}\_{j=t}^\infty}(s_t)$ defined with (ii) smaller than the regret $V^{\star}(s_t) - V^{\pi_t}(s_t)$ defined with (i), so it can be seen that the regret with (ii) is a more lenient metric than (i), which means an algorithm that can achieve certain guarantee under regret (i) should also be able to achieve the same guarantee under regret (ii). Thus, this means our , and the lower bound under our regret (i) is at least the one for (ii). But overall, this difference is not big enough to change the order of regret bounds, and we included [14] in our table for completeness. We can be clearer about the relationship between these two definitions and its implications when we revise the paper.
>
> >  The improvement would have been marginal even if the objectives considered were the same, and when taking into account it may not even be an actual improvement --- the contribution seems questionable.
>
> > The improvement is rather incremental, shaving off a factor of $O(SA)$ ...
>
> Our burn-in improvement can have a huge effect when the state-action space is immense, which is what usually happens in practice and what this paper is focused on. We will revise our paper to make this limitation clearer. It is also important to note that the burn-in cost is not the only contribution we make; our algorithm is also the first model-free algorithm to achieve minimax regret optimality $\widetilde{O}(\sqrt{\frac{SAT}{(1-\gamma)^3}})$. Compared to the state-of-the-art result in [14], we reduced the space complexity from $O(S^2A)$ to $O(SA)$ and the computational complexity from $O(S^2AT)$ to $O(T)$.
>
> > It seems like the approach given here borrowed most of the algorithm from the finite horizon case [54] ...
>
> > In addition, the algorithm is complicated and presentation does not make it particularly easy to understand.
>
> We indeed used the reference-advantage idea from [54] and was clear about this in Section 3. To be precise, there is a difference between how we control the reference update $V^{\mathrm{R}}$, as our approach uses an additional lower confidence bound estimate to control the reference better. On the other hand, we are unsure how the slow-switching technique could “reduce the infinite horizon problem to a finite horizon one”. Our switching follows an adaptive schedule rather than a fixed one, so it doesn’t seem anything can be reduced to a finite horizon. In addition, our algorithm switches slowly, at a rate of once every $O(\sqrt{T})$ steps on average, and it would not make sense to consider a reduction to a finite-horizon MDP with horizon $O(\sqrt{T})$.
>
> Let us also underscore the nontrivial difference between the finite-horizon setting and infinite-horizon discounted setting when it comes to *online* regret minimization. These two settings give rise to different problem structures, which has been long recognized in the existing literature and also been briefly pointed out in our introduction. In the infinite-horizon setting, we optimize regret over one single trajectory, and there is always dependence between any pair of time steps along the trajectory. Such dependence can cause statistical difficulties and give rise to an infinitely expansive structure in the error decomposition. In contrast, the finite-horizon setting resets the trajectory every $H$ steps, so any statistical dependence lasts for at most $H$ steps. Neither of the two settings can be reduced to the other under online regret minimization.
>
> For this reason, there is little resemblance between our analysis and the one in [54]. If we apply the analysis from [54] naively to the infinite-horizon setting, it will not converge. The two proofs diverge from the very beginning: while [54] manipulate the regret into a clipped pseudo-regret and expand it for further decomposition, we decompose the regret directly and cope with the infinite expansion problem (unique to finite-horizon) with a recursion-style algebraic manipulation (Appendix D).
>
> As for the presentation of the algorithm, while the technicality might be involved, everything can be summarized into three key ideas: UCB, variance reduction and the adaptive slow-switching technique, which are detailed in Section 3. We will be happy to clarify if you have any specific question about the algorithm.
>
> Please let us know if you have any further questions and we will be happy to answer. Thanks again!

---

> > ### Author Response · Authors · 2023-08-18
> >
> > Dear Reviewer zms9,
> >
> > Thank you again for taking the time to review our paper! You asked some very important questions in your review that we really hope to resolve. Could you please let us know if our rebuttal has addressed them sufficiently? Please don’t hesitate to contact us if you have any further questions.

---

> > > ### Comment · Reviewer_zms9 · 2023-08-18
> > >
> > > I would like to thank the authors for their rebuttal, and I appreciate the thoughtful and detailed reply. I would also like to thank them for the  kind reminder.
> > >
> > > Some of my concerns have been addressed, however my main concerns remain, in summary:
> > > 1) It is not clear the result improves over prior work, since there is no formal treatment of the comparison between the two objectives.
> > > 2) It is not clear the rate obtained is optimal, since there is no formal lower bound for the regret objective considered in this work.
> > > 3) The improvement in the regret (if we do compare to prior work) is not high impact, as it is an $SA$ additive factor in the regret bound. *I respect the authors opinion, that they perceive this as a significant improvement, but I don't share their view.*
> > >
> > > In addition:
> > > * I agree that the computational and space complexities are additional valuable contributions.
> > >
> > > Regarding your reply to my first concern:
> > > > Thus, this means our , and the lower bound under our regret (i) is at least the one for (ii)
> > > * I understand the intuition you are trying to convey, but this is not a formal argument. It remains unclear if one metric is stronger than the other or if these are incomparable. Do you not agree?
> > > * Don't you think it is problematic to present your results along with previous upper bounds and lower bounds in the same table, even though these are bounds for different objective functions?

---

> > > > ### Author Response · Authors · 2023-08-18
> > > >
> > > > We thank the reviewer for the reply and the detailed explanation about the main concern. We would like to make the following clarifications:
> > > >
> > > > 1. We included the existing results under the other regret definition in our table (along with the existing results that use the same definition as ours) because we felt they are relevant and wanted to be as comprehensive as possible. We have made clear in our current submission about this difference between the two objectives (for instance in Section 2).
> > > >
> > > >     We respect the reviewer’s concern. If the reviewer strongly believes the results with these two objectives should not be discussed together, we can be clearer about this definition difference and remove any direct comparison between results with different objectives when we revise the paper. Specifically, we can present the results with the other regret definition as a different line of work and move them from Table 1 to a separate section for discussion.
> > > >
> > > >     However, we do believe our study of the regret defined in Equation (3) is meaningful and important. We don’t think it is fair to say this result has no improvement over prior works just because there is no formal conversion from our metric to the other metric. But to address your concern, we can be clearer that our result makes improvements in this specific type of regret when we revise the paper.
> > > >
> > > > 2. We also want to emphasize that our contributions in reducing the computational complexity and space complexity are important and no less significant than our contribution in the $SA$ reduction in the burn-in cost. We respectfully disagree with the reviewer and believe these contributions of our paper are significant.
> > > >
> > > > 3. Let us also be clear that our result is minimax-optimal under the regret definition we used. Although we believe the relation between the two regret definitions is clear, we can provide a lower bound for the regret definition we used. It can be obtained with a simple modification of a classic lower bound construction like the one in Azar et al. (2013). Due to the space limit, we didn’t include it in our current submission, and we apologize for any misunderstanding it has caused. Let us present the construction below. We will also include this lower bound and its full proof in our revised paper and hope the reviewer could kindly reassess after reading this lower bound. Please also let us know if you have any questions.
> > > >
> > > > Consider the MDP $(\mathcal{S}, \mathcal{A}, \gamma, r, P)$ with
> > > > $$P(s | 0, a) = \gamma \mathbf{1}(s = 0) + (1-\gamma)\mu(s), ~ r(0, a) = 0, ~ \forall s, a;$$
> > > > $$P(s' | s, a) = p_{s, a} \mathbf{1}(s' = s) + (1-p_{s, a})\mathbf{1}(s' = 0), ~ r(s, a) = 1, ~ \forall s > 0, a, s',$$
> > > > where we represent states and actions with natural numbers, and the state distribution is $\mu(s) = \frac{1}{S-1}\mathbf{1}(s > 0)$. Also, for every $s$, let $p_{s, a} = \gamma$ for some unknown $a$, which will be the optimal action at $s$, and $p_{s, a} = \gamma - c(1-\gamma)^2\varepsilon$ for the remaining $A-1$ actions for some small constant $c > 0$.
> > > >
> > > > This construction follows from the classic example for infinite-horizon discounted MDPs.
> > > > Then it can be easily checked that for $s > 0$,
> > > > \begin{align}
> > > > V(s) = \frac{1}{1 - 2\gamma^2/(1+\gamma)},~ V(0) = \frac{\gamma}{1+\gamma}V(s)
> > > > \text{  and  }
> > > > d^{\pi}(s) \gtrsim \frac{1}{S}, \text{ for any }\pi.
> > > > \end{align}
> > > > In addition, it is a classic result (from Fano's inequality) that we need at least $\Omega(\frac{A}{(1-\gamma)^3\varepsilon^2})$ samples for any state $s > 0$ to find an $O(\varepsilon)$-optimal policy (which is the optimal policy). Combining this with the fact that the occupancy measure for every state $s > 0$ is on the order of $\frac{1}{S}$, we can bring everything into the regret definition for the final lower bound of $\Omega(\sqrt{\frac{SAT}{(1-\gamma)^3}})$.

---

> > > > > ### Comment · Area_Chair_9cuR · 2023-08-19
> > > > > **Different regret measures**
> > > > >
> > > > > Dear authors,
> > > > >
> > > > > Following your discussion with reviewer zms9, I would appreciate your response to the following questions:
> > > > > 1. Could you further elaborate on your reasons for choosing to work with the current regret measure? Was there any specific motivation where this measure is more natural or was it due to technical reasons? While the intuition you gave in your response is clear, it seems as if the other definition is more 'grounded', in the sense that it measures the suboptimality of the real policy that will be played.
> > > > > 2. How tailored is your analysis to the regret measure you chose? Would working with the other regret measure require substantial algorithmic/proof changes?
> > > > > 3. Just to make sure - would you confirm that for the sake of this discussion, there is currently no formal connection between the two measures (e.g., your bound implies a bound on the other measure, up to reasonable additive/multiplicative constants)?
> > > > >
> > > > > Thanks in advance,
> > > > >
> > > > > Area Chair 9cuR

---

> > > > > > ### Author Response · Authors · 2023-08-20
> > > > > >
> > > > > > Dear Area Chair 9cuR,
> > > > > >
> > > > > > We really appreciate your attention to our discussion in this thread. Regarding your questions:
> > > > > >
> > > > > > 1. Thank you for this question. We actually think the regret definition we used is more natural than the other one (which is the reason why we chose it). Recall that the goal of RL is to learn a (near-)optimal policy, and normally, people solve the RL problem because they need a near-optimal policy to use somewhere else. In our paper, they can just take an output policy $\pi_t$ from our algorithm. In contrast, they would need to take $\\{\pi\_j\\}\_\{j=t\}^\infty$ in the other setting, because the optimality guarantee is on $\\{\pi\_j\\}\_\{j=t\}^\infty$ under the other metric. This can be difficult to use as a “policy”. Our $\pi_t$ is just a stationary mapping $\mathcal{S} \to \mathcal{A}$ that can be readily used, but $\\{\pi\_j\\}\_\{j=t\}^\infty$ needs to be computed dynamically as it is being used. In addition, since there always exists a deterministic, **stationary** optimal policy in the infinite-horizon discounted setting, the metric we chose aligns better with this setting.
> > > > > >
> > > > > >     It is also important to note that both metrics are intended to measure “regret” during the period in which an algorithm is executed for $T$ steps. In this case, it does not make sense to consider what happens beyond step $T$, but the other regret, defined as $\sum_{t=1}^T V^{\star}(s_t) - V^{\\{\pi_j\\}_{j=t}^\infty}(s_t)$, not only depends on $\pi_t$ for $t > T$ but also needs $\pi_t$ as $t$ does to $\infty$, which does not make sense either, since both regrets are intended for the finite-time regime only. But ultimately, there isn’t one definition that is formally better than the other between these two. What we have discussed here is only our motivation or rationale for choosing this regret definition we used.
> > > > > >
> > > > > >     On this note, we would also like to point out that although this difference in stationary vs non-stationary metrics has existed across a number of infinite-horizon papers for years, it has never been particularly discussed in the existing literature. In fact, these works just compare results with different definitions directly. For example, Table 1 in [14] directly compares the regret guarantee for their algorithm (under the non-stationary regret definition) with the guarantees for UCB-multistage and UCB-multistage-adv (under the stationary regret definition like ours) with no formal or informal discussion or conversion; Table 1 in [54] directly compares its guarantee under the stationary metric with results from other works obtained under the non-stationary metric like Wang et al. (2020) without noting this difference. Such direct comparison appears to be the norm of this body of literature, as this difference is not expected to change the minimax lower bound of the problem or the upper bound of an algorithm, and this is why we didn’t underscore this difference when we wrote the initial version of this paper. Conversely, it might not be completely fair to hold this paper to a different standard from the existing literature. We hope the area chair could also kindly take this into consideration when evaluating this paper.
> > > > > >
> > > > > >     Overall, as we have expressed in our previous comments, we respect the concerns raised by Reviewer zms9 and the mathematical rigor he or she demands. We are willing to revise our paper to present our result differently by clearly differentiating the results with different metrics, but the majority of our paper—the algorithm, theorem and proof—is sound and ready.
> > > > > >
> > > > > > Wang et al. (2020): Y. Wang, K. Dong, X. Chen, and L. Wang. Q-learning with UCB exploration is sample efficient for infinite-horizon MDP. In 8th International Conference on Learning Representations, ICLR 2020, Addis Ababa, Ethiopia, April 26-30, 2020.
> > > > > >
> > > > > > 2. We think our approach should be able to achieve similar optimality under the other regret metric too, since the algorithm in [14] also has mechanisms to control the variance of its estimates (our variance reduction and slow-switching techniques should be able to function in the same way in an analysis for the other metric), but our analysis is long and it is difficult for us to say this rigorously at this point.
> > > > > >
> > > > > > 3. To the best of our knowledge, there is no formal (or informal) discussion about this in the existing literature. As aforementioned, the existing literature ignores this difference and compares results across definitions directly.
> > > > > >
> > > > > > Thank you again for your attention and time! Please let me know if you have any other questions.
> > > > > >
> > > > > > Paper 2623 Authors

---

> > > > > > ### Author Response · Authors · 2023-08-21
> > > > > >
> > > > > > Dear Area Chair 9cuR,
> > > > > >
> > > > > > Following our last reply to your question #3, we just want to add that we were just able to prove a formal connection between the two regret metrics in the case that the algorithm is low-switching like ours. It is detailed in our latest reply to Reviewer zms9 and can prove the regret of our algorithm is also on the same order under the other metric. This should be able to resolve any disagreement in this thread and enable a comparison between our result and the results proved with the other metric. Thank you for your attention!

---

> > > > > > > ### Comment · Reviewer_zms9 · 2023-08-21
> > > > > > >
> > > > > > > Thank you, this is very helpful.
> > > > > > >
> > > > > > > Given this, and the combination of improvements (regret additive factor, time and space complexities), I will raise my score.

---

> > > > > > > > ### Author Response · Authors · 2023-08-21
> > > > > > > >
> > > > > > > > Dear Reviewer zms9,
> > > > > > > >
> > > > > > > > We really appreciate your reply and your dedication to the review process. We will include these additional bounds and lemmas we made during this thread when we revise our paper for a more rigorous comparison with related results.

---

> > > > > > > > > ### Comment · Area_Chair_9cuR · 2023-08-22
> > > > > > > > >
> > > > > > > > > Dear authors,
> > > > > > > > >
> > > > > > > > > Thanks for engaging in this discussion with Reviewer zms9. I am glad to hear that the issue of the different regret measures was resolved.
> > > > > > > > >
> > > > > > > > > Best regards,
> > > > > > > > >
> > > > > > > > > Area Chair 9cuR

---

> > > > > > > > > > ### Author Response · Authors · 2023-08-22
> > > > > > > > > >
> > > > > > > > > > Dear Area Chair 9cuR,
> > > > > > > > > >
> > > > > > > > > > Thank you for your reply and attention. We will be sure to revise the presentation of our paper accordingly and include the additional bounds and lemmas we made in this thread.
> > > > > > > > > >
> > > > > > > > > > Best regards,
> > > > > > > > > > Authors

---

> > > > > ### Comment · Reviewer_zms9 · 2023-08-19
> > > > >
> > > > > Thank you for your reply and clarification.
> > > > >
> > > > > Given this my concern #2 above has been addressed.
> > > > >
> > > > > 1. I agree previous upper bounds should be discussed in the context of your work, but the details of the comparison should be made clearer in the exposition and table, well before the remark in Section 2.2. If the table clearly states the bounds are for different objectives I see no harm in putting them in the same table.
> > > > > In addition, I agree the study of the regret definition used in the paper is important. However, as long as it is not compared formally to the notion studied in the prior works, I don't think we can say this work strictly improves over them. **Currently it seems to me we should judge this work as a study of different objective that is not directly comparable to previous upper bounds. I think it is still a valid contribution as such, but then the exposition should be revised, and the review process should focus on aspects other than the exact factors in the bounds** (which, again, cannot be considered actual improvements if the objective is different..).
> > > > >
> > > > >
> > > > > 2. I acknowledge (as I did in the previous reply) that the computational and space complexities are valid contributions.
> > > > > But as I wrote above it is not clear to me how we can say these *improve* over prior work.

---

> > > > > > ### Author Response · Authors · 2023-08-20
> > > > > >
> > > > > > Dear Reviewer zms9,
> > > > > >
> > > > > > Thank you for the reply and further clarifications on your view that these two regret definitions should be treated as two different lines of work. As we express in previous comments, we are willing to change the way we present our result. Finally, regarding your latest #2, let us quickly note that even when we do not compare our result with the ones using the other regret definition, it is still an improvement to the line of work that follows the same regret definition as we do, since there does not exist any regret-optimal result under this regret definition yet (not even a model-based optimal algorithm).

---

> > > > > > > ### Comment · Reviewer_zms9 · 2023-08-20
> > > > > > >
> > > > > > > Thank you for your response.
> > > > > > >
> > > > > > > Regarding your reply to the AC;
> > > > > > > > In fact, these works just compare results with different definitions directly. For example, Table 1 in [14] directly compares the regret guarantee for their algorithm (under the non-stationary regret definition) with the guarantees for UCB-multistage and UCB-multistage-adv (under the stationary regret definition like ours) with no formal or informal discussion or conversion; Table 1 in [54] directly compares its guarantee under the stationary metric ...
> > > > > > >
> > > > > > > First I would like to note that I don't think it is a valid line of argument, to point out the same sort of invalid comparison was done by prior work.
> > > > > > > But in addition, to my understanding the comparison done in [14] is completely formal.
> > > > > > > * [14] clearly state at the top of their Table 1 that the regret bounds were derived by converting the sample-complexity guarantees of all algorithms (excluding Double Q-learning, which is a different story).
> > > > > > > They refer the reader to Appendix A.1, where they formally show, using a standard "explore-exploit" type argument, how they convert a sample complexity guarantee to a regret bound as defined in their work. This argument applies equally well regardless of the type of sample complexity guarantee (be it that of [55] or that of [21]); when using the algorithm of [55], in the exploit phase you just stick with the stationary policy it outputted.
> > > > > > > There is nothing informal here, everything is proved.
> > > > > > >
> > > > > > > Regarding Table 1 in [54];
> > > > > > > * There are no tables in the arxiv version of that paper. In addition they consider a finite horizon, episodic setup. Which paper (or version) did you mean to refer to?
> > > > > > >
> > > > > > > Please note I am not trying to be negative for spite. Please correct me if what I said above is incorrect or inaccurate. I would also like to note, again, that amending the exposition and asking for your work to be judged as one that does not directly improve prior work, seems much more reasonable to me than claiming the same "mistake" was done before you (if at all it was indeed).
> > > > > > >
> > > > > > >
> > > > > > > ### References
> > > > > > > I am using the numbering from the authors submission.
> > > > > > >
> > > > > > > [14] Jiafan He, Dongruo Zhou, and Quanquan Gu. Nearly minimax optimal reinforcement learning for discounted mdps. Advances in Neural Information Processing Systems, 34:22288–22300, 2021.
> > > > > > >
> > > > > > > [21] Tor Lattimore and Marcus Hutter. Pac bounds for discounted mdps. In International Conference on Algorithmic Learning Theory, pages 320–334. Springer, 2012.
> > > > > > >
> > > > > > > [54] Zihan Zhang, Yuan Zhou, and Xiangyang Ji. Almost optimal model-free reinforcement learn ingvia reference-advantage decomposition. Advances in Neural Information Processing Systems, 33:15198–15207, 2020.
> > > > > > >
> > > > > > > [55] Zihan Zhang, Yuan Zhou, and Xiangyang Ji. Model-free reinforcement learning: from clipped pseudo-regret to sample complexity. In International Conference on Machine Learning, pages 12653–12662. PMLR, 2021.

---

> > > > > > > > ### Author Response · Authors · 2023-08-20
> > > > > > > > **Reply to Reviewer zms9**
> > > > > > > >
> > > > > > > > Dear Reviewer zms9,
> > > > > > > >
> > > > > > > > Thank you for your reply and time.
> > > > > > > >
> > > > > > > > - You stated in your first bullet point:
> > > > > > > >
> > > > > > > > > This argument applies equally well regardless of the type of sample complexity guarantee (be it that of [55] or that of [21]); when using the algorithm of [55], in the exploit phase you just stick with the stationary policy it outputted.
> > > > > > > >
> > > > > > > > You’re correct that this argument can be applied to the result from [55] by keeping the stationary policies during the conversion. However, this converted regret would be in terms of stationary policies, and this regret definition is the same as the one we considered in our paper. In Table 1 of [14], this regret guarantee converted from [55] is directly compared with [14]’s own regret guarantee, which is based on the non-stationary regret definition $\sum_{t=1}^T V^{\star}(s_t) - V^{\\{\pi_j\\}_{j=t}^\infty}(s_t)$. Such direct comparisons between stationary and non-stationary metrics in the literature is exactly our point. Thus, **what you described here actually buttresses our previous argument**.
> > > > > > > >
> > > > > > > > In addition, **we do not think the conversion from sample complexity metric to regret in Appendix A.1 of [14] is “formal” as you claimed**. When they compare their regret guarantee with a sample complexity guarantee from another work, they obtained the converted regret by upper-bounding $\mathrm{Regret}(T)$ with its worst-case realization without violating the given sample complexity guarantee. This converted regret is very loose and cannot offer much information about how much regret the algorithm really incurs when compared against the result in [14], whose analysis is tailored to their regret definition.
> > > > > > > >
> > > > > > > > Let us be more specific. Recall a sample complexity of exploration guarantee is an upper bound on the total number of times $V^\star(s_t) - V^{\pi_t}(s_t) \ge \epsilon$. Denote the to-be-converted sample complexity guarantee with $N(\epsilon, \delta)$, where $\delta$ is the standard failure probability threshold for high probability results. [14] obtains the converted regret as follows:
> > > > > > > > $$\qquad\qquad \sum_{t=1}^T V^\star(s_t) - V^{\pi_t}(s_t)$$
> > > > > > > > $$\le \sum_{t\in[T],~ V^\star(s_t) - V^{\pi_t}(s_t) \ge \epsilon} V^\star(s_t) - V^{\pi_t}(s_t) + \sum_{t\in[T],~ V^\star(s_t) - V^{\pi_t}(s_t) < \epsilon} V^\star(s_t) - V^{\pi_t}(s_t)$$
> > > > > > > > $$\le \frac{N(\epsilon, \delta)}{1-\gamma} + T\epsilon.$$
> > > > > > > > In words, they simply take an upper bound of $\epsilon$ for $V^\star(s_t) - V^{\pi_t}(s_t)$ during the time steps when $V^\star(s_t) - V^{\pi_t}(s_t) < \epsilon$ and take an upper bound of $\frac{1}{1-\gamma}$ during the time steps when $V^\star(s_t) - V^{\pi_t}(s_t) \ge \epsilon$. They also upper-bound the total number of times $V^\star(s_t) - V^{\pi_t}(s_t) < \epsilon$ with $T$.

---

> > > > > > > > > ### Author Response · Authors · 2023-08-20
> > > > > > > > > **Reply to Reviewer zms9 (continued)**
> > > > > > > > >
> > > > > > > > > The point is, such looseness makes a comparison between a regret guarantee converted in this way and a regret guarantee tailored to the regret metric quite vacuous, and this seems no more rigorous than a direct comparison between a regret with stationary policies and a regret with non-stationary policies, which is also done in [14]. In fact, the authors of [14] have clarified that this conversion/comparison is not formal in the very same section:
> > > > > > > > >
> > > > > > > > > > Though both metrics have been used to describe the performance of an algorithm, these two metrics are not directly comparable. More specifically, algorithms with fewer but larger sub-optimalities will have a small sample complexity of exploration but a high regret. In contrast, algorithms with a lot of moderate sub-optimalities will have a high sample complexity of exploration but a low regret.
> > > > > > > > >
> > > > > > > > > Hence, if you actually consider a comparison resulted from this kind of conversion formal, we could easily make a similarly “formal” conversion from the non-stationary regret in [14] to our stationary regret, by finding the worst-case regret upper bound under our metric conditioned on the guarantee in [14]. But we don’t think such a comparison is formal or reveals much insight.
> > > > > > > > >
> > > > > > > > > Overall, we respect you for your exactness, as we have expressed in a few prior comments, and we are willing to differentiate results with different regret metrics more clearly, if you truly believe results under different metrics offer no contributions or revelations to each other, but at the same time, it might come across as double-standard to deem the direct comparison between stationary and non-stationary metrics in [14] acceptable and find a loose conversion/comparison that the authors themselves consider informal “completely formal”. Also, please let us note again that even when we exclude the results that follow the non-stationary metric from the comparison, there are still existing results developed under the stationary metric, and it is not correct to say our result “does not directly improve prior work”. Lastly, since we all build our own work more or less on the foundation of prior art as researchers, it is probably a little impulsive to call a convention evidenced in multiple works from a body of literature a “mistake” when one personally disagrees with it.
> > > > > > > > >
> > > > > > > > > - Regarding your second bullet point, it should be [55] instead of [54], and the table is in the appendix. Sorry about this typo.

---

> > > > > > > > > > ### Comment · Reviewer_zms9 · 2023-08-21
> > > > > > > > > >
> > > > > > > > > > It is unfortunate we cannot agree on such basic facts.
> > > > > > > > > >
> > > > > > > > > > Please let me know if you disagree with the following: **[14] shows that if you run an explore-exploit type conversion on the algorithm of [55], you will get the regret bound mentioned in their table, according to the regret definition in their paper**. (As you mention, it **also** obtains the regret guarantee w.r.t. your definition.)
> > > > > > > > > >
> > > > > > > > > > More specifically with regards to the points you have made:
> > > > > > > > > > > In Table 1 of [14], this regret guarantee converted from [55] is directly compared with [14]’s own regret guarantee, which is based on the non-stationary regret .... Such direct comparisons between stationary and non-stationary metrics in the literature is exactly our point. Thus, **what you described here actually buttresses our previous argument**.
> > > > > > > > > >
> > > > > > > > > > This is incorrect. When playing a fixed stationary policy, as is done in the exploit phase of the conversion of the algorithm of [55], the loss on all rounds (in the exploit phase) is the same regardless of which definition of regret you use.
> > > > > > > > > >
> > > > > > > > > > > In addition, **we do not think the conversion from sample complexity metric to regret in Appendix A.1 of [14] is “formal” as you claimed**. When they compare their regret guarantee with a sample complexity guarantee from another work, they obtained the converted regret by upper-bounding Regret(�) with its worst-case realization without violating the given sample complexity guarantee. This converted regret is very loose
> > > > > > > > > >
> > > > > > > > > > > The point is, such looseness makes a comparison between a regret guarantee converted in this way and a regret guarantee tailored to the regret metric quite vacuous, and this seems no more rigorous than
> > > > > > > > > >
> > > > > > > > > > Your thoughts are incorrect - this is not a matter of opinion. Formal has a very simple meaning; there is a mathematical proof (along with a well defined algorithmic procedure) that shows how to obtain the stated regret bounds, w.r.t.~the same definition used.
> > > > > > > > > >
> > > > > > > > > > You are correct that the actual regret bound may be better, but we currently have no proof for this. A comparison of this sort is still completely formal. In addition, it is fair, as long as the authors did not overlook a simple argument that shows a stronger bound. This type of comparison is standard practice, and in particular the type of sample complexity -to- regret conversion is also a standard argument.
> > > > > > > > > >
> > > > > > > > > > > Hence, if you actually consider a comparison resulted from this kind of conversion formal, we could easily make a similarly “formal” conversion from the non-stationary regret in [14] to our stationary regret, by finding the worst-case regret upper bound under our metric conditioned on the guarantee in [14]. But we don’t think such a comparison is formal or reveals much insight.
> > > > > > > > > >
> > > > > > > > > > If you can argue (formally) something along these lines in your work - by all means you should do it. This would then give the readers an apples to apples comparison. If not, you should not directly compare, but only state other results for context, while making it clear they are for a different objective function.
> > > > > > > > > >
> > > > > > > > > > > In fact, the authors of [14] have clarified that this conversion/comparison is not formal in the very same section:
> > > > > > > > > > > > Though both metrics have been used to describe the performance of an algorithm ...
> > > > > > > > > >
> > > > > > > > > > [14] is referring to sample complexity vs regret, not stationary regret vs non-stationary regret.
> > > > > > > > > >
> > > > > > > > > >
> > > > > > > > > > ### Comment on Stationary Regret
> > > > > > > > > > I would like to mention that this discussion along with reading more of the references, has highlighted it makes little sense to consider the stationary regret definition as in your work.
> > > > > > > > > > It does indeed make much sense to consider it as a sample complexity measure, since it is completely reasonable to ask for an algorithm to output a stationary policy that is close to optimal. This is the metric considered by [55], they do not considered stationary regret.
> > > > > > > > > > For this reason their Table 1 is, like that of [14], **completely formal**. They are comparing sample complexities, not regret. The metric considered is sample complexity of reaching an $\epsilon$-optimal policy, be it stationary or not. Here, non-stationary sample complexity is simply a larger class. Unfortunately the same does not apply in case of stationary vs non-stationary regret.
> > > > > > > > > >
> > > > > > > > > > Non stationary regret makes sense because it measures the loss incurred by the actual policy played by the algorithm. It is not clear what is the rational in stationary regret, w.r.t. measuring online performance. I do still understand it's validity as a performance metric, just not as one that measures loss incurred during learning, which is (arguably) the primary goal of a regret measure.
> > > > > > > > > >
> > > > > > > > > > The only other work you mention that considers stationary regret is [19]. This paper has not been published anywhere.

---

> > > > > > > > > > > ### Comment · Reviewer_zms9 · 2023-08-21
> > > > > > > > > > >
> > > > > > > > > > > Just a small correction;
> > > > > > > > > > >
> > > > > > > > > > > > In fact, the authors of [14] have clarified that this conversion/comparison is not formal in the very same section:
> > > > > > > > > > > > > Though both metrics have been used to describe the performance of an algorithm ...
> > > > > > > > > > >
> > > > > > > > > > > Indeed [14] is referring to sample complexity vs regret, not stationary regret vs non-stationary regret, but that is not the point.
> > > > > > > > > > > The point is they are saying the two metrics are not directly comparable, not that their comparison is informal. Since they are not directly comparable, the authors provide a formal argument that converts the sample complexity guarantee to a regret guarantee.

---

> > > > > > > > > > > ### Author Response · Authors · 2023-08-21
> > > > > > > > > > >
> > > > > > > > > > > It is indeed unfortunate that we still cannot reach an agreement at this point. In hope of resolving this dispute completely, let us provide a mathematical proof of the closeness between the stationary regret and non-stationary regret, conditioned on the fact that $\{\pi_t\}$ is low-switching, which is true for our algorithm. (Note that this relation might not hold in general; we only prove this when the algorithm is low-switching.)
> > > > > > > > > > >
> > > > > > > > > > > Let us first prove a lemma as follows: for any algorithm that switches at most $|\mathcal{T}|$ times during its execution over $T$ steps, the difference between the stationary regret and non-stationary regret is no more than $\sqrt{\frac{T\log^3 T}{(1-\gamma)^3}} + \frac{|\mathcal{T}|\log T}{(1-\gamma)^2}$.
> > > > > > > > > > >
> > > > > > > > > > > Specifically, let us define
> > > > > > > > > > > $$\mathcal{T} = \{1 \le t \le T : \pi_t \ne \pi_{t+1}\}$$
> > > > > > > > > > > and
> > > > > > > > > > > $$\mathcal{T}_H = \{1 \le t \le T : t + h \in \mathcal{T}\text{ for some }0 \le h \le H\}.$$
> > > > > > > > > > >
> > > > > > > > > > > Then we have
> > > > > > > > > > > $$\quad \bigg|\sum_{t \notin \mathcal{T}\_H} \Big(V^{\\{\pi\_j\\}\_{j=t}^\infty}(s_t) - \sum_{i = 0}^H \gamma^i r(s_{t+i}, a_{t+i})\Big)\bigg|$$
> > > > > > > > > > > $$\le \bigg|\sum_{t \notin \mathcal{T}\_H} \Big(\mathbb{E}\Big[\sum_{i = 0}^H \gamma^i r(s_{t+i}, a_{t+i})\Big] - \sum_{i = 0}^H \gamma^i r(s_{t+i}, a_{t+i})\Big)\bigg| + \frac{\gamma^HT}{1 - \gamma}$$
> > > > > > > > > > > $$\le \sum_{k = 1}^{H}\bigg|\sum_{t = jH + k \notin \mathcal{T}\_H} \Big(\mathbb{E}\Big[\sum_{i = 0}^H \gamma^i r(s_{t+i}, a_{t+i})\Big] - \sum_{i = 0}^H \gamma^i r(s_{t+i}, a_{t+i})\Big)\bigg| + \frac{1}{T}$$
> > > > > > > > > > > $$\lesssim \sqrt{\frac{T\log^3 T}{(1-\gamma)^3}},$$
> > > > > > > > > > > where the second line holds when $H \gtrsim \frac{\log T}{1-\gamma}$, and the last line makes use of the Azuma-Hoeffding inequality and is inspired by Appendix A.2 of [14].
> > > > > > > > > > >
> > > > > > > > > > > Similarly, we also have
> > > > > > > > > > > $$\bigg|\sum_{t \notin \mathcal{T}\_H} \Big(V^{\pi_t}(s_t) - \sum_{i = 0}^H \gamma^i r(s_{t+i}, a_{t+i})\Big)\bigg|
> > > > > > > > > > > \lesssim \sqrt{\frac{T\log^3 T}{(1-\gamma)^3}},$$
> > > > > > > > > > > since $\pi_t = \pi_{t+i}$ for $t \notin \mathcal{T}_H$.
> > > > > > > > > > >
> > > > > > > > > > > Putting these two relations together leads to
> > > > > > > > > > > $$\quad\big|\mathsf{Regret}\_{\text{ours}}(T) - \mathsf{Regret}\_{\text{other}}(T)\big|$$
> > > > > > > > > > > $$\le \Big|\sum_t \big(V^{\pi_t}(s_t) - V^{\\{\pi\_j\\}\_{j=t}^\infty}(s_t)\big)\Big|$$
> > > > > > > > > > > $$\lesssim \sqrt{\frac{T\log^3 T}{(1-\gamma)^3}} + \frac{|\mathcal{T}|\log T}{(1-\gamma)^2},$$
> > > > > > > > > > > by noticing that $\Big|\sum_{t \in \mathcal{T}_H} \big(V^{\pi_t}(s_t) - V^{\\{\pi\_j\\}\_{j=t}^\infty}(s_t)\big)\Big| \lesssim \frac{|\mathcal{T}_H|}{1-\gamma} \lesssim \frac{|\mathcal{T}|\log T}{(1-\gamma)^2}$.
> > > > > > > > > > >
> > > > > > > > > > > Given this lemma, we can see that if an algorithm’s switching cost satisfies $|\mathcal{T}|$ being $o(\sqrt{T})$ and its regret is $\widetilde{O}(\sqrt{\frac{SAT}{(1-\gamma)^3}})$ under either of the two regret metrics, we can say this algorithm will have the same order of regret under the other metric too. Specifically to our algorithm, this lemma allows us to conclude the following:
> > > > > > > > > > >
> > > > > > > > > > > We can use (130) in our analysis to bound the total number of switches in our algorithm, which gives $|\mathcal{T}|$ being $\widetilde{O}(\frac{S^{3/4}A^{3/4}T^{1/4}}{(1-\gamma)^{1/4}})$ for our algorithm. Plugging this $|\mathcal{T}|$ into the lemma above, it implies our algorithm also incurs $\widetilde{O}(\sqrt{\frac{SAT}{(1-\gamma)^3}})$ regret under the non-stationary regret metric, which matches the lower bound in [14].
> > > > > > > > > > >
> > > > > > > > > > > We can write this into a theorem and include this in our paper, along with the lemma above. Overall, this should be able to resolve any disagreement in this thread and enable a comparison between our result and the results proved with the other metric.

---

### Decision · Program_Chairs · 2023-09-21

**Decision:**

Accept (poster)

**Comment:**

The paper studies regret minimization in discounted infinite-horizon Markov Decision Processes.
The paper presents a minimax optimal algorithm for this setting that improves previous results from two different aspects:
1. Improved space complexity of $O(SA)$ and computational complexity of $O(T)$.
2. Improved additive constant in the regret bound, so that the regret would be dominated by the asymptotic rate from an earlier stage in the interaction ('short burn-in time').

After a long discussion with the authors about the regret measures used in the paper and their relation to previously used measures, all reviewers have agreed that the above contributions are sufficient for the paper to be accepted.

The authors are encouraged to modify the paper to include the relation between the different regret measures (as was promised during the discussions).